# Exotic tree plantations in the Chilean Coastal Range: Balancing effects of discrete disturbances, connectivity and a persistent drought on catchment erosion

Violeta Tolorza[1], Christian H Mohr[2], Mauricio Zambrano-Bigiarini[1,3], Benjamín Sotomayor[4], Dagoberto Poblete-Caballero[5], Sebastien Carretier[6], Mauricio Galleguillos[3,7], and Oscar Seguel[5]

[1]Universidad de La Frontera. 4811230 Temuco, Chile
[2]Institute of Environmental Science and Geography, University of Potsdam, Germany
[3]Center for Climate and Resilience Research (CR2). Blanco Encalada 2002, 4th floor, Santiago, Chile
[4]Dron Aerogeomática SpA, Spatial Data and Analysis in Aysén, Chile
[5]School of Agronomic Sciences, Universidad de Chile. 11315 Santiago, Chile
[6]Geosciences Environnement Toulouse, IRD, OMP, UPS, CNRS, Université de Toulouse, France
[7]Facultad de Ingeniería y Ciencias, Universidad Adolfo Ibáñez, Peñalolen, Chile

**Correspondence:** Violeta Tolorza (violeta.tolorza@ufrontera.cl)

**Abstract.** The Coastal Range in the Mediterranean segment of Chile is a soil mantled landscape with potential to store valuable freshwater supplies and support a biodiverse native forest. Nevertheless, human intervention has been increasing soil erosion for $\sim$200 yr, culminated by intense management of exotic tree plantations during the last $\sim$45 yr. At the same time, this landscape has been severely stressed by a prolonged megadrought. As a result, such combination of stressors complicates disentangling
the effects arising from anthropogenic disturbances and/or hydroclimatic trends on sediment fluxes at the catchment scale.

In this study we calculate decennial catchment erosion rates from suspended sediment loads and compare them with a millenial catchment denudation rate estimated from detrital $^{10}$Be. We then contrast both these rates against the effects of discrete anthropogenic disturbance events and hydroclimatic trends. Erosion/denudation rates are similar in magnitude on both time scales, i.e. 0.018 $\pm$0.005 mm/yr and 0.024 $\pm$0.004 mm/yr, respectively. Recent human-made disturbances include logging
operations during all seasons and a dense network of forestry roads, thus increasing structural sediment connectivity. Further disturbances include two widespread wildfires (2015 and 2017, respectively) and a Mw 8.8 earthquake in 2010.

We observe decreased suspended sediment loads during the wet seasons for the period 1986-2018 coinciding with declining streamflow, baseflow and rainfall. The low millenial denudation rate agrees with a landscape dominated by slow diffusive soil creep. However, the low decennial erosion rate and the decrease in suspended sediments disagrees with the expected effect
of intense anthropogenic disturbances and increased structural (sediment) connectivity. Such paradox suggests that either suspended sediment loads and, thus, respective catchment erosion, are underestimated, and/or that decennial sediment detachment and transport were masked by decreasing rainfall and streamflow, thus weakened hydroclimatic drivers. Our findings indicate that human-made disturbances and hydrologic trends may result in opposite, partially offsetting effects on recent erosion, yet both contributing to landscape degradation.

# 1 Introduction

Over 75% of Earth's ice-free land has been altered by humans (Ellis and Ramankutty, 2008), with severe consequences for sediment transport during the Anthropocene (Syvitski et al., 2022). Land Use and Land Cover Changes (LULCC) are important in increasing soil erosion (Borrelli et al., 2020). Human-made forests – or better, tree plantations (DellaSala, 2020) – are frequently disturbed by logging and the implementation of forestry roads. Such disturbances may intensify soil erosion (e.g., Schuller et al., 2013; Sidle and Ziegler, 2012), as may heavy machinery traffic (e.g., Malmer and Grip, 1990), wildfires and terracing (e.g., Martins et al., 2013). Short rotational cycles, i.e. the period between planting, harvesting, and replanting of tree plantations, also change hillslope stability by cycles of root strength decay and recovery, which in turn promote landsliding and debris flows (Imaizumi et al., 2008; Montgomery et al., 2000). Ultimately, all such processes may modify sediment trajectories and storage on hillslopes and along rivers (Wainwright et al., 2011) with long-lasting impacts on sediment yields for periods of 10-100 years (Moody and Martin, 2009; Bladon et al., 2014).

The Chilean Coastal Range (CCR) between 35-37.5º S is a landscape of gentle and largely convex hillslopes (Fig. 1) under Mediterranean conditions. Its morphology results from relatively slow denudation rates by soil creep on regolith-mantled landscapes (Roering et al., 2007), yet modified by the underlying bedrock (Gabet et al., 2021). Here, forests, soils and water are closely coupled (Galleguillos et al., 2021). Currently, the remnants of secondary native forests (i.e. successional forests growing in areas where forest cover was removed at some point in the past) stand on soils that are up to 2 m (Soto et al., 2019), suggesting that soils under natural vegetation cover could reach an even greater thickness. In the absence of snow storage, these soils form a major fresh water supply along the Mediterranean CCR, which many rural communities rely on. Thus, decision-making regarding land management is strategic for the resilience of these communities (e.g., Gimeno et al., 2022), especially under recent (Garreaud et al., 2020) and projected (IPCC, 2021) conditions of water scarcity.

The CCR has experienced deforestation for more than 200 years (Armesto et al., 2010) intensifying soil erosion, as has been recognized by Bianchi-Gundian (1947) and Chilean governments in the middle of 20th century (IREN, 1965). From the beginning of 20th century, governments blamed environmental issues due to deforestation to promote the expansion of tree plantations (e.g., CONAF and MINAGRI, 2016; Pizarro et al., 2020). The most relevant transformation of land cover began with the law DL 701 (1974) to subsidize the forestry sector (Manuschevich, 2020). This law and following political action accelerated LULCC, which in practice transformed degraded lands, shrublands and native forest into industrially managed tree plantations (Heilmayr et al., 2016). From ∼450,000 ha of tree plantations in 1974 (Barros, 2018), their spatial extent increased to at least some 2.8 ±0.2 million ha in 2011 (Heilmayr et al., 2016), mostly within the Mediterranean CCR (Fig. 1).

In Chile, tree plantations are managed mostly as monocultures of fast-growing *Eucalyptus* spp or *Pinus Radiata* stands. For both species, the rotation cycles are as short as 9-12 and 18-25 years, respectively (INFOR, 2004; Gerding, 1991). Harvesting commonly occurs by clear-cutting, and their extent usually expand over entire hillslopes (Fig. 2, supplementary video S1). Such practice is permitted by current Chilean law, as clear-cutting requires environmental impact assessments only for harvest areas $>=$ 500 ha/yr or $>=$ 1,000 ha/yr in Mediterranean and Temperate regions, respectively (*Artículo Primero, Título I,*

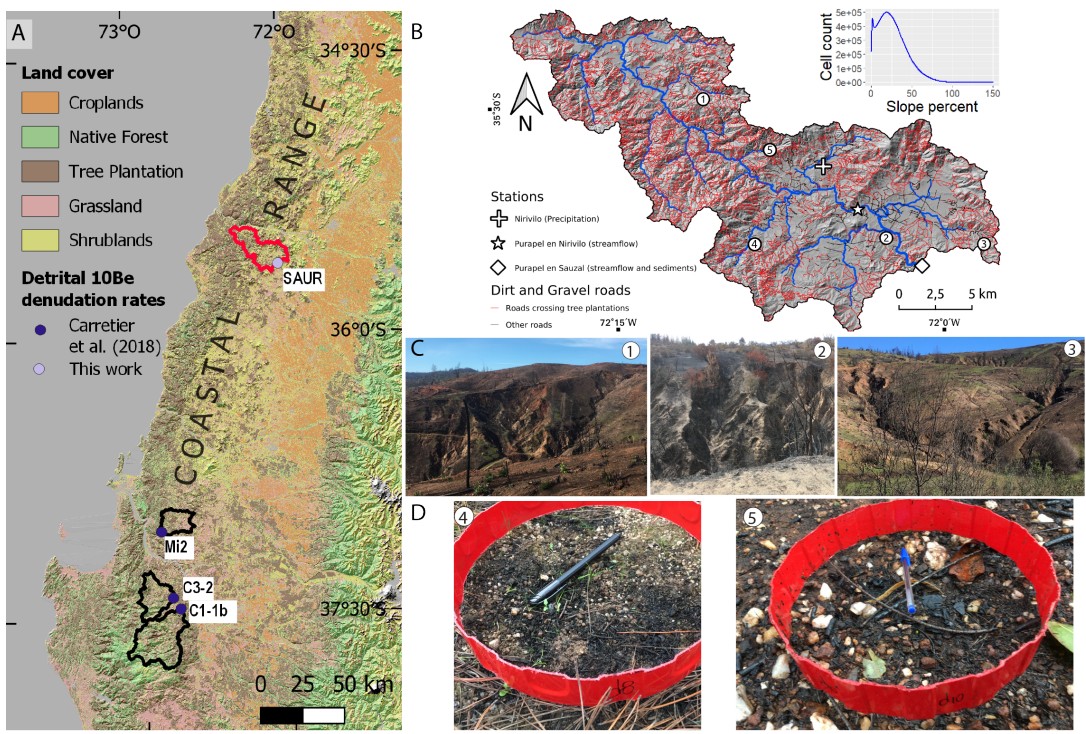

**Figure 1.** Study region. A. Land cover in the Coastal Range (Zhao et al., 2016) and catchments with published detrital [10]Be denudation rates outlined in black (Carretier et al., 2018). The Purapel catchment, which denudation rate is presented in this work, is represented in red. B. Purapel catchment. All the detected forestry roads intersecting tree plantations and the position of photos in C and D are shown. Elevation data comes from a 5-m resolution LiDAR DTM obtained in 2009. C. Photos captured on hillslopes of Purapel catchment after the fire. D. Photos of bare topsoil after the fire on hillslopes with (4) granitic and (5) metamorphic lithologies.

*Artículo 3, m.1* at Chilean Law 19.300, 2013). As a consequence, the CCR ranks among regions with highest forest loss and gain worldwide (Hansen et al., 2013).

Tree plantations are frequently intersected by dense networks of logging roads. These roads facilitate access and use of heavy forest machinery, storage and transport of timber, as well as the subsequent (re-)plantation. Logged hillslopes, like logging roads, are important sediment sources and routes during storms and after wet-season clear-cutting (Schuller et al., 2013, 2021; Aburto et al., 2020). This is not surprising, since they remain bare and prone to compaction by heavy machinery transit. Post-harvest erosion is mainly rainfall triggered (Aburto et al., 2020; Schuller et al., 2013) after exceeding soil-hydrologically

specific rainfall intensity thresholds (Mohr et al., 2013). The erosional work efficacy depends on the logging season, which is higher for wet season logging (Mohr et al., 2014). Nevertheless, post-harvest erosion may last more than one season. Indeed, Aburto et al. (2020) reported highest post-harvest soil loss in a catchment sustaining a one-year-old plantation.

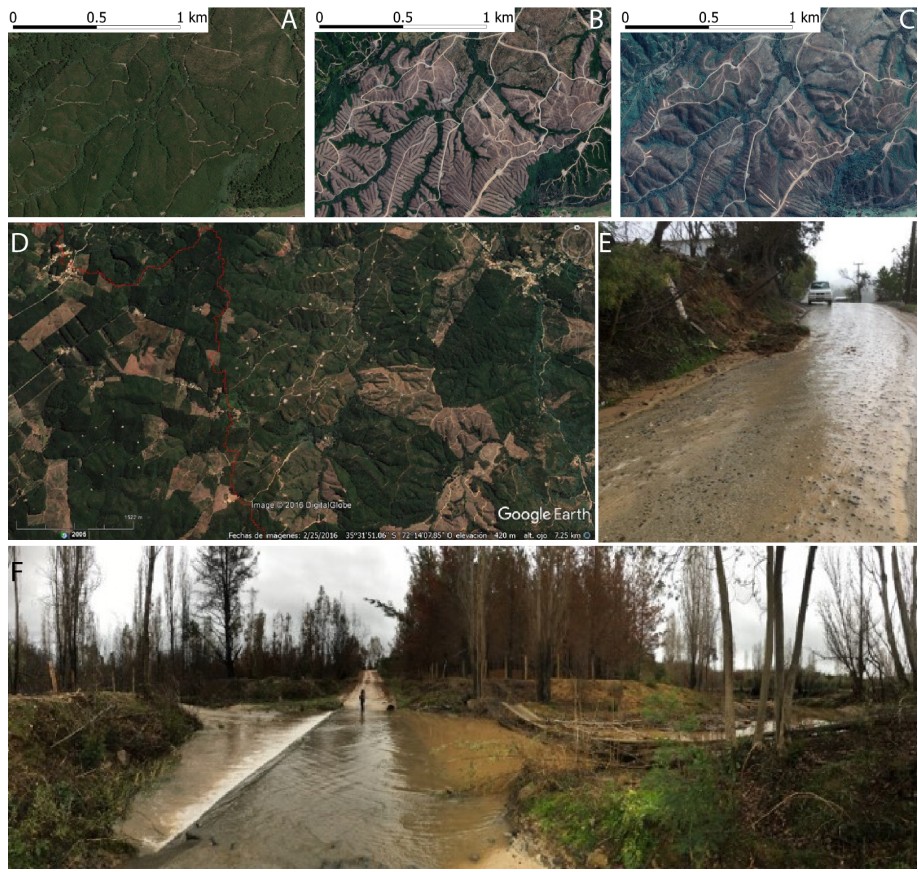

**Figure 2.** Forest roads in the Purapel catchment under different stages of the tree plantation rotational cycle and their connection to streams. A-D Google images (©Google Maps 2016, ©Google Maps 2021, ©Google Maps 2022, ©Google Earth 2016, respectively), E-F pictures of a gravel road and its connection to a stream during a storm in July 2017.

Within the plantation cycle, roads are prime sources and routers of sediments in catchments covered by tree plantations (Schuller et al., 2013). These roads often intersect streams, which form bypasses to preferentially route sediment (Fig. 2), increasing the efficacy of mass transfer within a geomorphic system, which is also called sediment connectivity (Wohl et al., 2019). In this case, road networks modify the pathways of runoff and sediments, and may also modify thresholds of rainfall to trigger sediment detachment and transport (for example, due to soil compaction), potentially affecting both the structural and functional components of sediment connectivity, as defined by Wainwright et al. (2011). This shift is also relevant to constraining off-site impacts of soil erosion (Boardman et al., 2019).

Despite the increase in structural connectivity, sediment mobilization depends mostly on specific thresholds of rainfall. For example, (functional) hydrologic connectivity to initiate runoff in recently logged areas required a threshold of 20 mm/hr in rainfall simulations on tree plantations near Nacimiento (Mohr et al., 2013). In the absence of long term records of rainfall intensities, hydro climatic trends in rainfall and streamflow are relevant to context catchment erosion. In Central Chile (30-

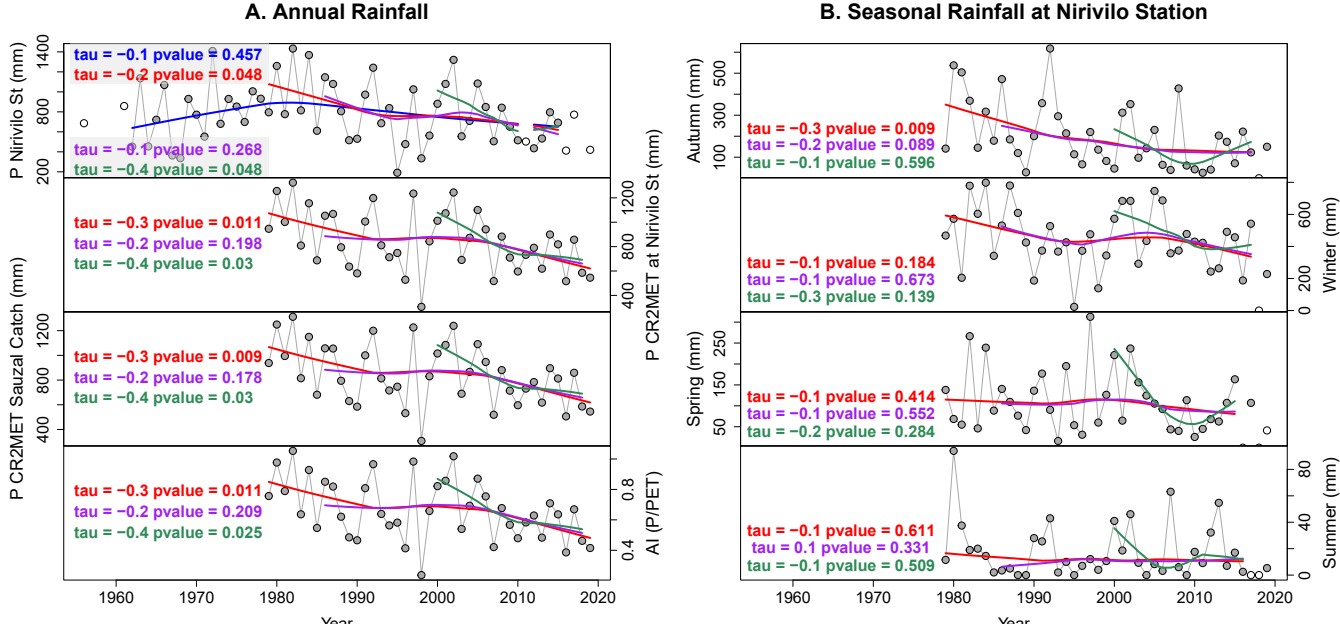

**Figure 3.** Annual and seasonal rainfall and annual aridity index (AI=P/PET) at Purapel catchment. Main monotonic trends are tested with Mann-Kendall and LOWESS smoothing for 1962-2015 (blue), 1979-2019 (red), 1986-2018 (purple) and 2000-2018 (green). Unfilled circles are discarded data. A. Annual rainfall and AI time series. B. Seasonal time series for Nirivilo station.

39ºS), rainfall decreased at a rate of approx. 4% per decade between 1960 and 2016 (Boisier et al., 2018), culminating in an unprecedented megadrought starting 2010 (Garreaud et al., 2020).

While the erosional response of logging is largely indisputable, hydrologic responses to tree harvest are ambiguous. On the one hand, logging may increase streamflow discharge, particularly for peak flows (Iroumé et al., 2006). On the other hand, logging may also decrease streamflow discharge due to enhanced groundwater recharge immediately after logging (Mohr, 2013). The distinct responses may most likely vary with tree species and age, harvest size, forestry treatment (thinning, clear cutting, replanting), riparian buffer width, and especially, with soil water storage decrease under recent drought conditions, which exacerbated declines in runoff (Iroumé et al., 2021).

Superimposed on the megadrought, recent increase in both magnitude and frequency in wildfire affects relatively more tree plantations compared to alternative land cover (Bowman et al., 2019). This is likely because fuel is more abundant under dense plantation cover, linking extensive and uninterrupted stretches of the landscape. Contrarily, native species distribute more patchiness (Gómez-González et al., 2017, 2018).

While the observed disturbances of the vegetation cover commonly increase soil loss (Aburto et al., 2020) and fluvial sediment yields (e.g., Reneau et al., 2007; Brown and Krygier, 1971), the long and persistent decline in rainfall (Méndez-Freire et al., 2022; Tolorza et al., 2019) together with the high water demands of tree plantations is expected to reduce water avail-

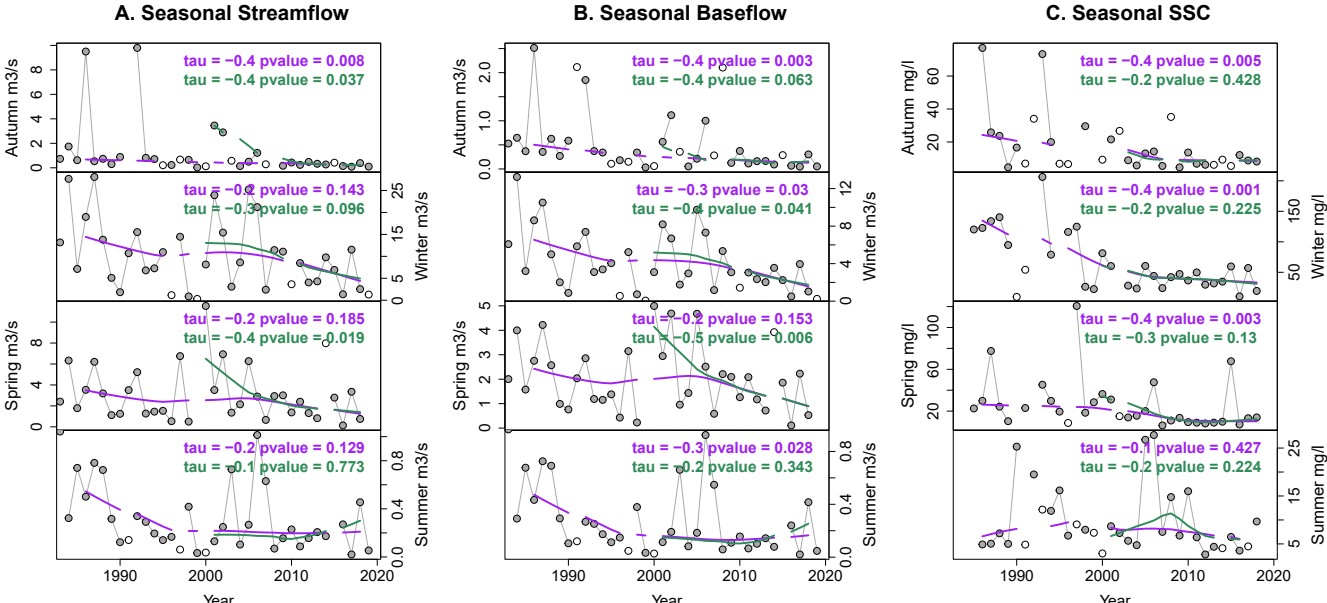

**Figure 4.** Mean seasonal streamflow, baseflow and suspended sediment concentrations at "Purapel en Sauzal" station on an annual basis. Main monotonic trends are tested with Mann-Kendall and LOWESS smoothing for 1986-2018 (purple) and 2000-2018 (green). Unfilled circles are discarded data. A. Mean seasonal streamflow at "Purapel en Sauzal" station. B. Mean seasonal baseflow at "Purapel en Sauzal" station. C. Mean seasonal suspended sediment concentration at "Purapel en Sauzal" station.

ability for both sediment detachment and subsequent mobilization. In most fluvial catchments, the long-term ($10^3 - 10^4$ years)
denudation rates exceed short-term rates (Covault et al., 2013). This picture, however, may flip vice versa if anthropogenic
soil erosion is high (Hewawasam et al., 2003; Vanacker et al., 2007). To evaluate this conundrum, we explore the catchment
scale erosion and denudation of the Purapel river. To this end, we establish a long-term benchmark based on detrital $^{10}$Be
denudation rate to compare recent sediment yields against. We include also discrete disturbance events, i.e. wildfires in 2017
and 2015, and wet-seasonal logging, and (dynamic) sediment connectivity associated with forestry roads. In a recent analysis
of this river, Pizarro et al. (2023) concluded that the concomitant afforestation and the decrease in sediment export out of the
catchment may form a causal relationship. Here, we analyze a wide range of meteorological and vegetational indexes over a
42-year period. We show that causes different from afforestation may explain the observed decline in sediment discharge, in
particular the drought that this catchment has been experiencing starting back in 2010.

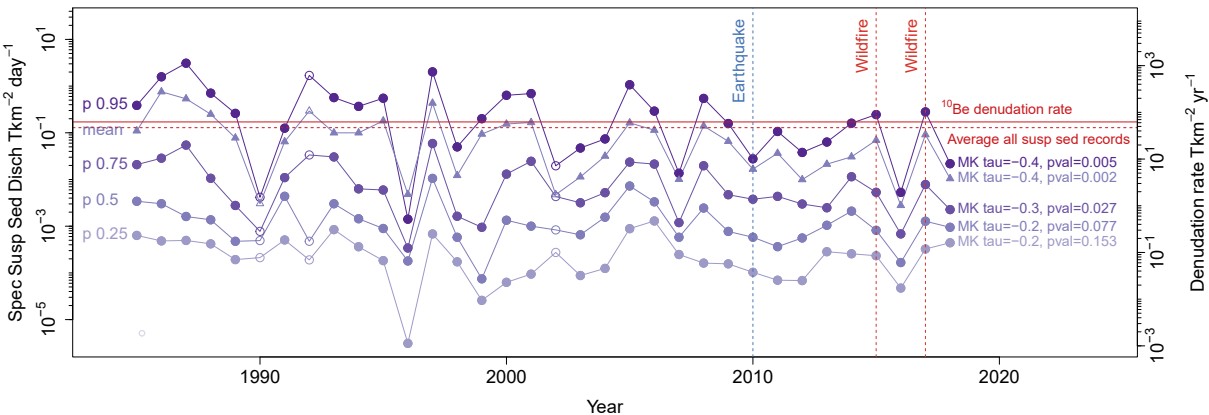

**Figure 5.** Denudation and specific $SSD$ at "Purapel en Sauzal" gauge. Distributions of $SSD$ for individual hydrologic years (March to Feb). Purple circles show percentiles (0.25, 0.5, 0.75 and 0.95) and purple triangles show the mean of daily data within each hydrologic year. Filled symbols represent years with more than 185 daily data. Catchment erosion/denudation rates are indicated in red. Solid line is the sediment yield equivalent to the $^{10}$Be denudation rate, dashed line is the average of all suspended sediment records. They are expressed both in daily and annual scale to facilitate comparisons.

**Table 1.** Published and new detrital $^{10}$Be denudation rates in the Mediterranean CCR. Denudation rates and their uncertainties were calculated with procedures described in Carretier et al. (2018). Characteristic time refers to the quartz residence time within a particle mean free path in rocks of 60 cm, and represent a timescale for steadily erosion (Lal, 1991). Details of $^{10}$Be concentrations and blanks analyzed in the CEREGE laboratory are available in the supplemental table S1.

| Name | Denudat. rate (mm/yr) | Denudat. rate unc. (mm/yr) | Char. time (kyr) | Lat | Lon | Catch. area (km$^2$) | Analyzed grain size (mm) | [$^{10}$Be] (at/g) | [$^{10}$Be] unc. (at/g) | Standard material | Source |
|---|---|---|---|---|---|---|---|---|---|---|---|
| SAUR | 0.024 | 0.004 | 25 | -35.6197 | -72.0171 | 406 | [0.5,1] | 143751 | 5469 | STD-11 | This work |
| Mi2 | 0.037 | 0.006 | 16 | -37.0488 | -72.8614 | 235 | [0.5,1] | 93772 | 4280 | 4325 | Carretier et al. (2018) |
| C3-2 | 0.039 | 0.007 | 15 | -37.4052 | -72.7976 | 357 | [0.5,1] | 97896 | 8272 | 4325 | Carretier et al. (2018) |
| C1-1b | 0.041 | 0.010 | 14 | -37.4652 | -72.7495 | 739 | [0.5,1] | 113680 | 20735 | 4325 | Carretier et al. (2018) |

## 2 Materials and methods

### 2.1 The Purapel catchment

The Purapel river drains the eastern flank of the CCR. The climate is of Mediterranean type. Mean annual rainfall is 845 mm, mean minimum and maximum air temperatures are 7.2 and 20.3ºC, respectively. The hydrological regime is exclusively pluvial bound (Álvarez-Garreton et al., 2018). The catchment is 406 km$^2$ and dominated by metamorphic (48%) and granitic (44%) lithologies. Elevation ranges between 164 and 747 m a.s.l. Most hillslopes are gentle (hillslope gradients around 16%), largely convex, and incised by gullies that converted this landscape into badlands (Fig. 1C). CIREN (2021) classified most of

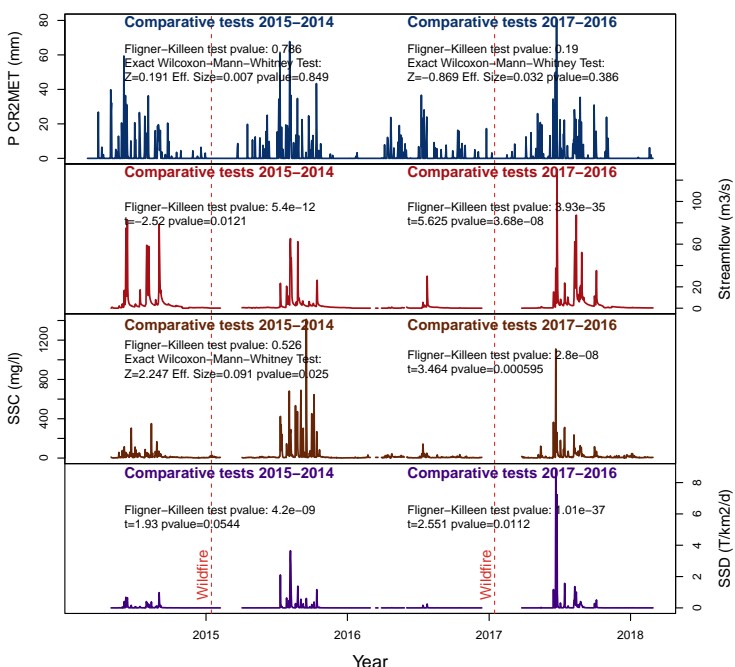

**Figure 6.** Daily hydrometric data for pre- and post- fire hydrologic years for the two large wildfire events. Rainfall, $Q$, $SSC$ and $SSD$ of pre- and post-fire hydrologic years were contrasted in homoscedasticity with the Fligner-Killeen test using an $\alpha =0.05$. Heterocedastic distributions were compared with Welch t-test. Homocedastic distributions were compared with Wilcoxon-Mann-Whitney test.

those hillslopes as severely affected by soil erosion. The dominant soil types are Inceptisols and Alfisols (Bonilla and Johnson, 2012). Soil properties are highly variable in space. Soils under tree plantations are generally thinner, more depleted in soil organic matter and with lower invertebrate diversity than soils under native forest (Cifuentes-Croquevielle et al., 2020; Soto et al., 2019). Throughout the entire soil column, the soil bulk densities of Eucalyptus ($1.38 \pm 0.08$ to $1.58 \pm 0.12$ g/cm$^3$) and Pine ($1.28 \pm 0.18$ to $1.53 \pm 0.13$ g/cm$^3$) stands are 40-80 % higher compared to native forests ($0.89 \pm 0.27$ to $1.25 \pm 0.24$ g/cm$^3$) (Soto et al., 2019). After the burning of mulch layer following the 2017 wildfire, bared surfaces exposed coarse topsoils (Fig. 1D).

## 2.2 Hydrometeorologic data and analysis

We downloaded daily rainfall and potential evapotranspiration PET data (1979-2020) for the Purapel catchment from CAMELS-CL dataset (https://camels.cr2.cl/). Daily rainfall data are derived from the CR2METv2 precipitation product, which merges the ERA5 reanalysis and local topographic data calibrated by an updated national rain-gauge network (DGA, 2017). In addition, we downloaded daily rainfall time series (1956-2019) at the Nirivilo rain gauge from Mawüm (https://mawun.cr2.cl/).

We analyzed trends of annual and seasonal rainfall for several periods using both data sets. We calculated annual aridity trends using the Aridity Index (AI), computed as the annual rainfall divided by annual potential evapotranspiration (AI=P/PET) using the CAMELS-CL data set. For annual analysis we considered years with more than 330 daily data and months with more than 27 daily data. For seasonal analysis ($\sim$90 days) we included data for seasons with more than 60 daily data - that is more than $\sim$66.6% of data for an assumed three month season. We defined seasons as Autumn (MAM), Winter (JJA), Spring (SON), and Summer (DJF). For the pre- and post-fire hydrologic years (Mar-Feb) we analyzed daily rainfall too. To this end, we used the gap-free gridded CR2METv2 product.

Streamflow and suspended sediment data are available in the Chilean General Directorate of Water (DGA) site (https://snia. mop.gob.cl/BNAConsultas/reportes). The DGA estimated daily streamflow discharge ($Q$) from single gauge stage readings using calibrated rating curves. Roughly once a month, the rating curves are updated by manual current meter measurements. The suspended sediment concentrations ($SSC$) were sampled on a daily-basis, too. All samples were obtained close to the water surface in the vicinity of the water stage. The samples were filtered using a cotton linter cellulose paper with 80% of collection efficiency for particles larger than 0.3 $\mu m$ (Advantec Qualitative Filter Papers 2, written communication from DGA operator). Then, they were dried, and burnt for 2 hours at 550-600ºC in DGA laboratories (Solar, 1999).

We calculated daily suspended sediment discharge ($SSD$, t/day) as the product of $Q$ (m³/s) and $SSC$ (mg/l), assuming those measurements as representative of the entire day (Pepin et al., 2010). This assumption is based in the catchment size and measurements of streamflow velocities we had performed in this landscape (supplemental material S1). In addition, we calculated the number of data, the percentiles and the mean value of $SSD$ for single hydrologic years. We calculated the daily baseflow at Purapel en Sauzal station with the Lyne and Hollick filter (Ladson et al., 2013), which is a standard approach used in several studies (e.g. Li et al., 2022; Huang et al., 2021; Teutschbein et al., 2015; Zhang et al., 2017). For the baseflow separation we used several $\alpha$ values between 0.5 and 0.95 and $n.reflected$=30 days as parameters.

The $Q$ and $SSC$ time series contain gaps, which are not seasonally clustered. In particular, gaps during the dry season are mostly related to ceased $Q$ (personal communication from DGA operator). We calculated the mean value and the number of daily $Q$, baseflow and $SSC$ data in each month and season. Based on the number of data (supplemental material S2), we discarded the monthly analysis of trends. Then, we calculated seasonal trends on streamflows and suspended sediments for seasons with more than 60 data points. All trends are computed with the Mann-Kendall test (Helsel et al., 2020) and plotted with an additional LOWESS smoothing (Cleveland, 1981). In addition, we computed quantiles and averages of $SSD$ for hydrologic years with more than 185 data points.

We also examined the existence of differences between the daily data for the hydrologic years before and after the fire, i.e., 2014 and 2016 versus 2015 and 2017, respectively (n>262 for each group of data). Given the non-normal distribution of all hydrological datasets, we used the Wilcoxon-Mann-Whitney test instead of the traditional t-test. That test requires the two compared populations have the same shape and variance (homoscedasticity). We used the non-parametric Fligner-Killeen test to judge whether our samples were homoscedastic (p-value larger than 0.05) or not (p-value lower than 0.05). If the compared populations had unequal shape or variance (heteroscedasticity), we used the Welch t-test (Skovlund and Fenstad, 2001).

### 2.3 Catchment-wide erosion and denudation rates

We obtained catchment-wide erosion rates for Purapel river at the gauge "Río Purapel en Sauzal" using two approaches for different time scales, short-term (decennial) from suspended sediments and long-term ($10^3$ to $10^4$ yrs) from detrital $^{10}$Be. A limitation of our approach is the fact that detrital $^{10}$Be rates include physical erosion and chemical weathering rates (von Blanckenburg and Willenbring, 2014), while suspended sediment yields account only for physical erosion of very fine sediment (Summerfield and Hulton, 1994), which excludes bedload and dissolved load. Thus, we regard our short-term erosion rates as minimum rates for landscape lowering.

For the short-term, we calculated the mean specific $SSD$ (t/km$^2$/yr) as the average of all records (06/1985 to 11/2018) on a yearly scale and normalized by catchment area (Pepin et al., 2010). We estimated resulting erosion rate (mm/year) assuming a mean soil bulk density of 2.6 g/cm$^3$ (Carretier et al., 2018).

For the long-term, we assume the $^{10}$Be concentrations within fluvial sands are proportional for catchment-wide averaged denudation rate (von Blanckenburg, 2005; Granger and Schaller, 2014). This rate integrates over a characteristic timescale that is inversely proportional to the denudation rate. These timescales are commonly longer than $10^3$ years (Covault et al., 2013). We therefore regard the $^{10}$Be derived rates as a reference that largely excludes recent human disturbances but includes low frequency and high magnitude erosion events (Kirchner et al., 2001; Carretier et al., 2013). We obtained a bulk sample of fluvial sands from the active river bed along a cross section close to the water stage "Río Purapel en Sauzal", collecting sands from the surface at three locations within ~10 m distance. We mixed all samples and sieved to a grain size fraction 0.5-1 mm.

The mixed sand sample was processed at the French AMS ASTER facility in CEREGE. In order to convert the $^{10}$Be concentration $C$ into catchment mean denudation rate, we neglect radioactive decay and assume steady state of $^{10}$Be concentration, leading to the classical following equation

$$\epsilon = \frac{1}{\rho C} P_{SLHL}(f_{sp} S_{sp} \Lambda_{sp} + f_{sm} S_{sm} \Lambda_{sm} + f_{fm} S_{fm} \Lambda_{fm}) \tag{1}$$

where $P_{SLHL} = 4$ at/g/yr is the sea-level-high-latitude total production rate of the considered nuclide (Martin et al., 2017). $f_{sp}$, $f_{sm}$ and $f_{fm}$ are the fractions of this production rate due to spallation, slow muons capture and fast muons averaged over the catchment area, respectively (Braucher et al., 2011). $S_{sp}$, $S_{sm}$, $S_{fm}$ are scaling factors depending on latitude and elevation averaged over the catchment area (Stone, 2000), and $\rho = 2.6$ g/cm$^3$. No geometric shielding correction for topography was applied (horizon < 20° in all directions). The uncertainty in the denudation rate is the propagation of the analytical uncertainty and an assumed uncertainty of 15% in the production rate.

### 2.4 Land cover changes

The Purapel catchment has experienced high rates of LULCC since the 19th century. This was largely due to the extensive increase in wheat production caused by the gold rushes in California and Australia (Cortés et al., 2022). Later on, between 1955 and 2014 tree plantations increased from (a minimum of) 10.27 (Hermosilla-Palma et al., 2021) to 203.5 km$^2$ (Zhao

et al., 2016). Recently, two large wildfires burned the catchment: In 2015 14% of the catchment area burned. In 2017 almost the entire catchment burned (95%) (Tolorza et al., 2022).

To describe recent LULCC in this catchment, we use land cover maps both from compiled sources (1955, 1975 and 2017) and from our own (1986, 2000, 2005, 2010 and 2015):

- The 1955 and 1975 land cover maps of Hermosilla-Palma et al. (2021) cover the headwaters of the Purapel catchment (157 km$^2$). These maps were made interpreting the land cover from the 1:70,000 aerial photograph (Hycon flight) for 1955, and from the 60 m resolution Landsat-2 MMS and the 1:30,000 aerial photographs of 1978 (CH-30 flight) for 1975.

- We used Landsat Surface Reflectance products to identify land cover classes during dry seasons of 1986, 2000, 2005, 2010 and 2015. We classified unburned land cover using the Maximum Likelihood Classifier (Chuvieco, 2008) which we trained and validated with 20 and 10 polygons for each class, respectively. We validated the results with field observations during 2014-2015. We sub-classified burned surfaces into low, moderate and severe fire according to the differences in NBR index of pre- and post- fire images (thresholds <0.1 - 0.269>, <0.27 - 0.659>, <0.66 - 1.3>, Key and Benson, 2006).

- The Land cover map of 2017 was made by Tolorza et al. (2022) with pre-fire Sentinel and LiDAR data. Here, this classification was resampled to 30 m resolution, to be compatible with LANDSAT classifications.

## 2.5 Logging roads and sediment connectivity

To identify changes in the structural connectivity we applied the Connectivity Index ($IC$, dimensionless) using the weighting factor ($W$, dimensionless) of Cavalli et al. (2013). $IC$ is a semi-quantitative approach to describe the degree of coupling between hillslopes and a target (for example, the stream network):

$$IC = \log_{10} \left( \frac{\overline{W}\,\overline{S}\sqrt{A}}{\sum_i \frac{d_i}{W_i S_i}} \right) \tag{2}$$

, where $\overline{W}$ and $\overline{S}$(m/m) are the average weighting factor and slope gradients on the upslope contributing area ($A$, m$^2$), respectively. $d_i$ (m), $W_i$ (dimensionless) and $S_i$ (m/m) are the path length, the weighting factor and the slope gradient on the $i$th cell in downslope towards a target.

$W$ is calculated from a DTM to account for the effect of topographic roughness. The Roughness Index ($RI$, m) is the standard deviation of the residual topography. The residual topography refers to the difference between the original DTM and

a smoothed version obtained by averaging DTM values on a $5 \times 5$ (=25) cell moving window:

$$RI = \sqrt{\frac{\sum\limits_{i}^{25}(x_i - x_m)^2}{25}} \qquad (3)$$

, where $x_i$ (m) is the value of one specific cell of the residual topography within the moving window, and $x_m$ (m) is the mean of all 25 window cells. The weighting factor is calculated as:

$$W = 1 - \frac{RI}{RI_{max}} \qquad (4)$$

, where $RI_{max}$ is the maximum value of $RI$ in the study area.

The routing of sediments on hillslopes is likely to change where a new temporary sink is closer downslope. Both the streams and the forestry roads may behave as temporary sinks of sediments after their detachment on hillslopes (Schuller et al., 2013, 2021), and this behavior can be addressed using those elements (the streams and the forestry roads) as targets for $IC$. We propose here an approach to estimate the related changes in sediment connectivity by means of the difference between two different targets:

$$RC = IC_{rs} - IC_s \qquad (5)$$

, where the subscripts of $IC$ refer to the target for its computation: $rs$ for the network formed by streams and roads and $s$ for the network formed only by the streams. We fed the model with a mapped forestry road network obtained from images available in the OpenLayers plugin of QGIS and post-2017-fire Sentinel compositions.

This approach accounts only for changes in sediment connectivity when forestry roads behave as sinks of sediments, lacking their effect as sources.

## 2.6 Disturbances in vegetation

We used the Breaks For Additive Season and Trend algorithm (BFAST, Verbesselt et al., 2010) on a LANDSAT collection to detect disturbances in vegetation at the pixel scale, i.e. $>= 30$m. In the Purapel catchment, disturbances $> 30$ m are mostly due to wildfires and/or clear-cuts. Such disturbances lean on the seasonal behavior of the NDVI index on a time series of LANDSAT surface reflectance (Level 2, Collection 2, Tier 1) for the period from 09/1999 to 10/2021. Clouds were filtered using the QA band which uses the CFMask algorithm (Foga et al., 2017). We used the same parameter set as Cabezas and Fassnacht (2018), namely the threshold value for disturbances set to 93 manually labeled reference polygons with fire events, clear-cuts and constant tree-cover. It is worth mentioning, that we applied a sieve filter to the results. Hence, only disturbances greater than 1 ha were considered. We trained the algorithm with the Landsat time series of 1999 to 2001. Given the disturbance regime of Purapel catchment (two large wildfires and possible loggings each 9 to 25 years) we run BFAST anticipating three possible breaks for the period 2002-2021. The accuracy assessment was performed on 35 manually drawn polygons that were randomly

distributed across the catchment. Since we were dealing with imbalanced data (looking for disturbances which we assume as anomalies in the time series), we used balanced accuracy and F1 score as the metrics to evaluate our data (Brodersen et al., 2010). Balanced accuracy was calculated as the arithmetic mean of sensitivity and specificity while F1 score was calculated as

$$F1score = 2 * [(precision * recall)/(precision + recall)] \tag{6}$$

Precision is the amount of correct positive predictions (true positives / (true positives + false positives)) and recall is how many positive predictions the model made over all positive cases (true positives / (true positives + false negatives)).

## 3 Results

### 3.1 Hydro climatic trends

At the annual scale at Nirivilo rainfall station, most data of the period 1962-2015 passed our completeness assessment criteria (53 of 54 years). In the case of CR2MET, the longest period analyzed here is 1979-2019. Judging Mann-Kendall tests and LOWESS smoothing, we did not find a single trend for the longest interval of records (1962-2015). Nevertheless, for the period after 1979, we obtained decreasing non-monotonic trends for rainfall, both at Nirivilo station and for the CR2MET product. That decrease is steeper for 2000-2019, but less pronounced for intermediate intervals such as 1986-2018. During 1986-2018, however, a decrease in seasonal rainfall is observed for Autumn, at the beginning of the hydrologic year (Fig. 3). Generally, the AI follows similar decreasing trends as is the case for rainfall, thus indicating persistently dry conditions across this catchment. For only 2 years (1982 and 2002) the AI was higher than 1. During all other years, potential evapotranspiration was greater than rainfall.

Streamflow data is available at Purapel en Nirivilo between 1979 and 2019 and at Purapel en Sauzal between 1981-2019. For Purapel en Sauzal streamflow data, results of baseflow separation are in the supplement material S3. We selected the results obtained with $\alpha$=0.7 for further trend analysis, given the observed magnitudes and shape of the baseflow time series. For the Suspended Sediment Concentration data, the longest period is 1985-2018. The seasonal analysis for "Purapel en Sauzal" station is in Fig. 4. Although none of those time-series is monotonic, the sharp decrease in suspended sediment concentrations is clear for the three wetter seasons (Autumn, Winter and Spring).

### 3.2 Catchment-wide erosion and denudation rates

[10]Be denudation rate resulted in 0.024 $\pm$0.004 mm/yr (Table 1), assuming a soil particle density of 2.6 t/m$^3$. This rate translates into a specific sediment yield of $62.4 \pm 10.4$ t km$^{-2}$yr$^{-1}$. This rate integrates over a characteristic timescale of ~25 kyrs. Together with published data, detrital [10]Be denudation rates in the CCR point to between 0.02 and 0.05 mm/yr (table 1, Carretier et al., 2018).

Following Pepin et al. (2010) we calculated the mean specific $SSD$ for all the records between 1985 and 2018 and a 30% of error (Pepin et al., 2010). For the Purapel catchment we estimate $47 \pm 14.1$ tkm$^{-2}$yr$^{-1}$, equal to 0.018 $\pm$0.005 mm/yr, assuming the same soil bulk density. Both rates do not statistically differ (Fig. 5).

### 3.3 Decennial trends in $SSD$ and comparison of pre- and post-fire hydrometric data

On a decennial time-scale, we observe decreasing trends in mean and high (p95 and p75) annual values of $SSD$, judging from Mann-Kendall p-values for $\alpha$ =0.05 (Fig. 5). Since 2010, high and medium percentiles of $SSD$ are lower than most of the previous hydrologic years.

On a shorter time-scale, in Fig. 6 we compare pre- and post-fire hydrometric data for the 2015 and 2017 wildfires. Compared series resulted to be homoscedastic for rainfall in the two comparisons, and for $SSC$ in the comparison before and after the 2015 wildfire. No changes in median rainfall values can be interpreted from the large p-value of their Wilcoxon-Mann-Whitney tests. Before and after the 2015 wildfire, judging from p-values for $\alpha$ =0.05 on Welch t-test or Wilcoxon-Mann-Whitney tests, mean streamflow diminished, but median $SSC$ and mean $SSD$ increased. Before and after the 2017 wildfire, mean streamflow increased, as well as mean $SSC$ and mean $SSD$.

### 3.4 Recent land cover changes

We developed five land cover maps for the period 1986-2015. The overall classification accuracy ranged between 83% and 92%. We distinguished between tree plantations, native forests, shrublands and seasonal grasslands. Seasonal grasslands included bare surfaces, seasonal pasture and sparse vegetation. We also classified seasonal grasslands to separate recently logged areas (clear-cuts) from other poorly vegetated areas.

Fig. 7 shows that the upper catchment was covered by a minimum of 1,000 ha of tree plantations and 5,500 ha of shrublands in 1955 (Hermosilla-Palma et al., 2021). Between the 1980s and the beginning of 21st century, the most prominent change comprised the transition from seasonal grasslands and shrublands into tree plantations. The first two classes covered a minimum of 23,550 ha in 1986 and 13,050 ha on 2005. During the same period, tree plantations expanded from 8,090 to 20,980 ha. Between the wildfires of 2015 and 2017, seasonal grasslands and shrublands together expanded to ~20,300 ha (Fig. 7).

### 3.5 Landscape disturbances

The result of our mapped road network is illustrated in Fig. 1. Using this road network on a 5 m resolution LiDAR, we estimate some 18,000 ha of increased sediment connectivity ($RC$ >0), and 150 ha of decreased sediment connectivity ($RC$ <0) (Fig. 8). $RC$ values exceeding the 95-percentile (>3.12) are 1,986 ha. That area of high $RC$ is mostly located on hilltops: 1,966 ha with $RC$ > 3.12 (i.e. 99%) present upstream contributing area <1 ha. Particularly these topographic settings exceed an empirical threshold between high and low connectivity for a mountain catchment, i.e. -2.32 (Martini et al., 2022). In the Purapel catchment the area above this threshold increased from 1,120 to as much as 6,570 ha simply due to the dense road network. This quantification, however, is done with a digital terrain model of coarser resolution compared to the original study of Martini et al. (2022) (5 m vs 0.5 to 2.5 m). The few pixels resulting in negative $RC$ are located on first order channels omitted in the stream network, specifically where those channels intersect more than one road or stream (Fig. 8D).

Based on our BFAST modeling, we obtained monthly time series of disturbances for 2002-2019 that we aggregate to the seasonal scale. We achieved a confusion matrix with a balanced accuracy of 0.86 and a F1 score of 0.69. For the complete

period (Fig. 9A) 13,640 ha of the Purapel catchment (33.7%) experienced one break in the NDVI time series, 16,810 ha (41.5 %) showed two breaks and 5,010 ha (12%) presented three breaks. The undisturbed 12.8% included tree plantation stands that remained unlogged, and seasonal grasslands that remained poorly vegetated. Considering the seasonality (Fig. 9B), the periods with largest disturbed areas were the summers (dry season) of 2015 and 2017 due to the wildfires. Both wildfires were detected in ∼5,000 and 24,000 ha, respectively. The periods that follow in disturbed area were the Autumn (wet season) of 2002 (∼2,000 ha) and 2007 (1,910 ha). The largest surface disturbed during a Winter and a Spring were 770 ha each in 2006 and 2009, respectively.

Compared to the dNBR classification for 2017 (Tolorza et al., 2022), the BFAST results detected less burned areas for the 2017 wildfire (33,618 vs 24,299 ha). This difference could be explained by the better capabilities of the dNBR index to detect burned areas, since it is a dedicated method to classify burned surfaces based in the NBR index of a pre- and a post- fire image (Key and Benson, 2006), while the BFAST algorithm was applied here on the NDVI, which is an index more suitable to detect the density of vegetation, and thus more sensitive to clear cuts.

## 4    Discussion

### 4.1    Long and short term catchment erosion

Both [10]Be denudation rate and suspended sediment erosion rate are surprisingly similar (Fig. 5). Hence, the decennial average of suspended sediment may capture at least the effects of erosion events recorded with [10]Be on the long-term. Both rates are low for fluvial catchments between 100-1,000 km$^2$ (Covault et al., 2013). Yet, those rates are similar to 3 tributaries of the Biobío river draining the eastern CCR, which are between $0.037 \pm 0.006$ and $0.042 \pm 0.008$ mm/yr (Carretier et al., 2018).

The low [10]Be denudation rate agrees with a landscape dominated by slow soil creep with occasional mass wasting triggered by intense rainfall events and earthquakes. This long-term rate, however, may even be overestimated: after 200 years of intense soil erosion, deep saprolite layers with low concentrations of [10]Be are widely exposed at earth surface. Those hillslopes might be depleted in [10]Be, which may lead to overestimate the total denudation (Schmidt et al., 2016).

Short term erosion does not exceed the long term denudation, as in other highly human-disturbed catchments (Hewawasam et al., 2003; Vanacker et al., 2007). However, we regard the suspended-sediment decennial erosion rate as a very conservative estimate for recent catchment scale erosion for three reasons.

First, we argue that this estimate does not account for possible effects of sediment storage and other transient processes that affect soils and streams. For a certain time lapse, only a fraction of sediment detached within a catchment is transported to the outlet and, thus, captured in the sediment yield due to processes such as deposition and storage (Walling, 1983). The Sediment Delivery Ratio ($SDR$) describes that fraction, and use to be expressed as $SDR = SY/E$, where $SY$ is the average annual sediment yield per unit area and $E$ is the average annual erosion over that same area. Considering empirical data (Walling, 1983; Ferro and Minacapilli, 1995; Lu et al., 2006), a $SDR$ of 0.7 may be used as a conservative first approach for catchments larger than 100 km$^2$, although this value may be a couple of orders of magnitude lower. Using the mean specific $SSD$ as $SY$ in the previous expression, our estimation of decennial erosion at Purapel catchment increases to $67 \pm 20$ tkm$^{-2}$yr$^{-1}$ or 0.026

± 0.007 mm/yr, which is even closer to the long-term denudation rate. The decennial catchment erosion may be much higher if the $SDR$ is orders of magnitude lower. More reliable estimates of $SDR$ can be done using spatially distributed models,
such as the linear lumped series model of hillslope storage and channel storage of Lu et al. (2006). That model considers a theoretical relationship between sediment delivery and physical parameters such as particle size and rainfall duration.

Second, previous studies on rivers in the western Andes indicate that catchment-scale erosion from gauging data may be underestimated due to under-sampling of extreme events (Vanacker et al., 2020; Carretier et al., 2018). Given the numerous data gaps of both streamflow and suspended sediments, we cannot rule this out. Also, we don't have sub-daily or depth-integrated
measurements of sediment concentrations.

Third, suspended sediments do not record the effects of chemical weathering on denudation rates. This process seems to be relevant in the CCR: in the absence of spatially resolved data of regolith thickness, single observations suggest thick saprolite layers (at least) locally (Vázquez et al., 2016; Mohr et al., 2012; Krone et al., 2021), while weathering rates at plot scale ranges between 0 and ~240 t/km$^2$/a in other latitudes of the CCR (Schaller and Ehlers, 2022) (26, 29, 32 and 37ºS). Thus, depending
on the magnitude of mass loss due to chemical weathering, total denudation in the short term can be equal or even higher than the reported long-term denudation. Local quantitative estimates of chemical weathering and soil production rates would help to constrain the magnitude of the difference between long and short term denudation. Indeed, deep chemical weathering may also lead to underestimating long-term denudation rates with [10]Be. Such underestimation may occur if mass loss is high under the depth at which most cosmogenic nuclides are produced, as has been seen in tropical landscapes dominated by dissolution
(Campbell et al., 2022).

### 4.2 Disturbances and trends in suspended sediments

The Purapel river catchment has been a staging ground for rapid expansion of tree plantations and a number of disturbances during the period of suspended sediment monitoring. This landscape has been affected by clear-cuts, two widespread wildfires and one Mw 8.8 earthquake. The expansion of tree plantations was mostly at the expense of poorly vegetated surfaces (Fig. 7).
Yet, plantation management includes extensive logging operations (supplementary videos) – mostly during wet seasons (Fig. 9) – and the construction and maintaining of forestry roads used by heavy machinery.

The distribution and density of a forestry road network impact the routing of sediments, because the roads may act both as sources or sinks of the particles detached from hillslopes. Our results show changes in structural sediment connectivity on a 18,150 ha sized surface, where the roads act as target (i.e., behave as sinks of sediments). Most of the surface experienced an
increase in structural sediment connectivity and thus an increase in sediment routing capacity downslope (Fig. 8). Consequently, detached soil is now (relatively) well connected to local sinks, even from hilltops (surfaces with upstream contributing area <1 ha). In this specific position of the landscape the soil moisture is likely lowest and, consequently, we can assume the soil production rate is slower compared to the mid-slope or toe positions across the CCR (Schaller and Ehlers, 2022). Thus, hilltop soils may be more difficult to recover during human time scales.
The decennial trends in mean and high suspended sediment discharge (Fig. 5) and the mean $SSC$ during wet seasons (Fig. 4) are decreasing. Despite the disturbances, which cluster towards the end of our time series, the changes on post-disturbance

suspended sediments did not overprint this decennial trend. For example, after the 2015 wildfire we found an increase in sediment concentrations but a decrease in mean streamflow (Fig. 6). Hence, we argue that the elevated sediment concentrations reflect the erosional signal of the wildfire. Nevertheless, the described changes in pre- and post-fire suspended sediments are limited, since high and medium percentiles of suspended sediment discharge after the fire are lower than most of the hydrologic years prior to 2010, the beginning of the megadrought (Fig. 5). Assuming that the suspended sediment record is representative for sediment yields of the Purapel river, the disturbance regime contrasts the expected sediment mobilization as reported from other landscapes (e.g., Reneau et al., 2007; Brown and Krygier, 1971).

## 4.3  Water availability, suspended sediments and sediment storage

The arid conditions, i.e. the ratio between annual precipitation and evapotranspiration, indicate increasing water scarcity. Such increasing water scarcity is consistent with a decline in streamflow and baseflow during the wet seasons, mostly pronounced in the autumn. Besides the unprecedented drought starting in 2010, the high root water uptake by fast-growing tree plantations are likely key for the severe reduction in water availability. Indeed, both tree plantations and drought conditions may reduce groundwater recharge (Iroumé et al., 2021; Huber et al., 2010). Also the loss of soils due to erosion may further reduce the water-storage capacity (Ratta and Lal, 1998). Taken together, enhanced water scarcity due to the drought and the tree plantations may have reduced the groundwater storage. This is consistent with the observed negative trend in baseflow. Streamflow is largely composed of groundwater and as groundwater declines, less streamflow is available to perform geomorphic work, which may reduce sediment transport.

In the described conditions of drought and land use, the frequency of minimum rainfall intensity required to trigger runoff and soil erosion on hillslopes may also be declining. At the same time, as rainfall and direct runoff exert control over sediment fluxes at streams (Andermann et al., 2012; Tolorza et al., 2014), sediment mobilization under the current hydrological regime may likely remain low despite landscape disturbances. Although we observed an increase in structural sediment connectivity, it is therefore possible that sufficient water was unavailable to effectively establish sediment connectivity, thus leading to increased residence times of sediments stored within the catchment. Storage may occur in diverse sediment compartments, including hillslopes, forestry roads, tributary junctions, fans and flood plains.

Indeed, a model in post-fire sediment cascades indicates that, even if post-fire erosion may be severe in source areas, a substantial fraction of the detached sediments may (intermittently) remain stored within valleys with only moderate delivery to the river network (Murphy et al., 2019). A sustained decline in water available for streamflow generation in a system like that would likely respond with increasing residence times of sediments.

## 4.4  Sediments on hillslopes and streams over time

The biogeochemical processes impacting vegetation and soils alike, the amount and grain size distribution of sediments supplied by hillslopes, and the dynamic of sediment mobilization on streams are all closely related (Terweh et al., 2021). More than 200 years ago, the primary native forests probably grew on deep and carbon rich soils with surfaces probably much more enriched in fine sediments than the present topsoils. In contrast, the landscape-scale wildfire of 2017 revealed coarse saprolite

widely exposed at the earth surface (Fig. 1D). Roots of fallen trees digging up coarse sediments spread out over most hillslopes. If we assume that landscape lowering has denuded 2 m of soil in 200 years, physical erosion should present one or more stages of much higher rates in the past and after the beginning of intense deforestation. Rates of 10 mm/year are required to produce such denudation in 200 years, which exceed both our short and long term estimates by 3 orders of magnitude. While intense soil erosion occurs, the most likely evolution of grain size distribution on topsoils is a quick depletion in fine sediments and a relative enrichment in coarse and more weather resistant sediments. Together with deforestation and wildfires, this time-dependent coarsening of the inorganic component of topsoils would be described as a reduction in the supply of fine sediments from hillslopes to the streams, which may lead to a transition from transport to supply limited catchment erosion. Because vegetation and soils have been systematically disturbed, and given the high degree of coupling of those landscape elements and sediment mobilization at streams, it is unlikely that the grain size distribution on rivers remain constant. Since short-term and long-term erosion are comparable, we interpret that after a rapid response of erosion to deforestation, denudation rates decreased due to exhausted sediments. Hence, it is plausible that part of the observed decline in suspended sediment is related to changes in the grain size distribution of sediments detached from soils. In this interpretation, both the long- and the short-term erosion can be considered supply limited. The first due to the scarcity of landslides and the second due to the low speed of soil (and fine sediment) production.

## 4.5 Tree plantations, native forests and landscape degradation at CCR

The expansion of tree plantations has been proposed as a tool to mitigate soil erosion (CONAF and MINAGRI, 2016). Recently, plantations have been favored as a better solution to mitigate soil erosion compared to native forests for the same Purapel catchment (Pizarro et al., 2020, 2023). At Purapel catchment, a direct comparison between native forest and plantations cannot be achieved for the period 1986-2018, because the major land cover transitioned from poorly vegetated surfaces to tree plantations (Fig. 7). Nevertheless, we can discuss whether the observed land management is a suitable solution for soil erosion mitigation in the CCR. There is abundant evidence of accelerated soil erosion in Chilean tree plantations, such as truncated soil profiles in a eucalyptus stand at 36º37'S (Banfield et al., 2018), a fourfold increase in net soil loss under pine stands relative to native forest at Talcamavida (37º7'S) and Nacimiento (37º30'S) (Aburto et al., 2020) or changes in nutrient cycles and increased sedimentation rates in coastal lakes, such as Matanza (33º45'S, Fuentealba et al., 2020), Vichuquén (34ºS, Fuentealba et al., 2021), San Pedro (36º51', Cisternas et al., 2001), and Lanalhue (37ºS, Alaniz et al., 2021). Based on such strong empirical evidence along the CCR and our own results (Fig. 2, 7, 8 and 9), we argue that the observed ongoing forest management of tree plantations promotes soil erosion and landscape degradation. In addition, soils in tree plantations are depleted in carbon and nutrients (Soto et al., 2019; Banfield et al., 2018), and inhibit lower invertebrate diversity (Cifuentes-Croquevielle et al., 2020) compared to soils under native forest. As a result, C and N stocks are relatively lower in tree plantations up to deep soil compartments (>120 cm) (Crovo et al., 2021). Soil organic matter is a key component for soil formation (Bernhard et al., 2018). For that reason alone, native forests rather than exotic tree plantations are a more appropriate land cover to regenerate soils and reverse or, at least, decelerate 200 years of intense soil erosion. Indeed, the protection and conservation of natural vegetation has the strongest effect on improving soil quality after water erosion (Vanacker et al., 2022). Also, empirical restoration ex-

amples available show that the transition from former Eucalyptus plantations to native forest is valuable in terms of improving water availability (Lara et al., 2021).

## 5 Conclusions

The Purapel catchment, as other similar catchments along the CCR, denudates slowly on millenial timescales. The decennial averaged suspended sediment discharge is similar in magnitude, although likely underestimating total denudation. Then, depending on the magnitude of the unconstrained portion of the denudation, decennial lowering of earth surface may be equal to or even higher than the long-term average.

Suspended sediment transport decreased during the wet seasons between 1986 and 2018, which, at first glance, conflicts with the disturbances observed in vegetation, especially the intense and widespread wildfires. The decrease in several hydroclimatic variables and forcings, including baseflow and aridity, coincides with lower suspended sediment loads. We argue that the low range of recent suspended sediment results from the observed decline in water availability, thus limiting the detachment and transport of sediments. Or in other words: the drought offsets the effects of disturbances and higher connectivity. Without sufficient water, the residence times of sediments are long. The contribution of tree plantations to reduce erosion, if any, is more related to their impact in water availability than directly to soil protection. Complementing this explanation, the coarsening of topsoils due to persistent soil erosion would lead to a reduction in the supply of fine grain size to the streams.

For most of the landscape disturbances described in this work, we cannot unambiguously quantify the overall effect on sediment fluxes. Yet, sediment fluxes are more efficient during periods of high flows which correspond to wetter conditions. Consequently, the sediment stored in the valleys, highly rich in nutrients and carbon, can be re-suspended during higher discharge events, causing temporarily delayed off-site problems for several decades to come.

In the studied landscape (the CCR of the El Maule region) the surface lowering in the last three decades is similar to or higher than the long-term benchmark. Thus, we argue it may be considered high for this specific system. That conclusion and the documented effects of local tree plantations on soil organic carbon, soil density and soil biodiversity are clear indicators of a degrading landscape.

*Code availability.* R scripts used in this study for data analysis are accessible upon request by contacting Violeta Tolorza (violeta.tolorza@ufrontera.cl)

*Data availability.* Supplemental data-sets related to this submission are available at:
https://doi.org/10.5281/zenodo.6958544
https://doi.org/10.5281/zenodo.6974312
https://doi.org/10.5281/zenodo.7328071

*Video supplement.* Time-lapses showing disturbances in vegetation are available at:

https://doi.org/10.5446/62703

https://doi.org/10.5446/62704

*Author contributions.* VT: Conceptualization, funding acquisition, project administration, methodology, field work, validation, investigation, writing - original draft; CHM: conceptualization, methodology, investigation, writing - deep review and editing; MZB, SC, MG and OS: methodology, investigation, writing - review and editing, BS and DP: field work, validation, writing - review and editing

*Competing interests.* The contact author has declared that none of the authors has any competing interests.

*Acknowledgements.* This paper arises mostly from research funded by ANID-FONDECYT project 11190864 and the UFRO Postdoc grant VRIP20P001, and received contributions from ANID-FONDAP project 15110009 and DFG project 493703771. We thank Claudio Ramirez Bravo (DGA) for describing details of hydrometric monitoring in the Purapel River, and for providing the actual location of the gauge *Río Purapel en Sauzal*. We appreciate the contributions made by Amanda Schmidt, Thomas Hoffmann, Veerle Vanacker, Paul Zuckerman, and the anonymous reviewers who improved early versions of this article.

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

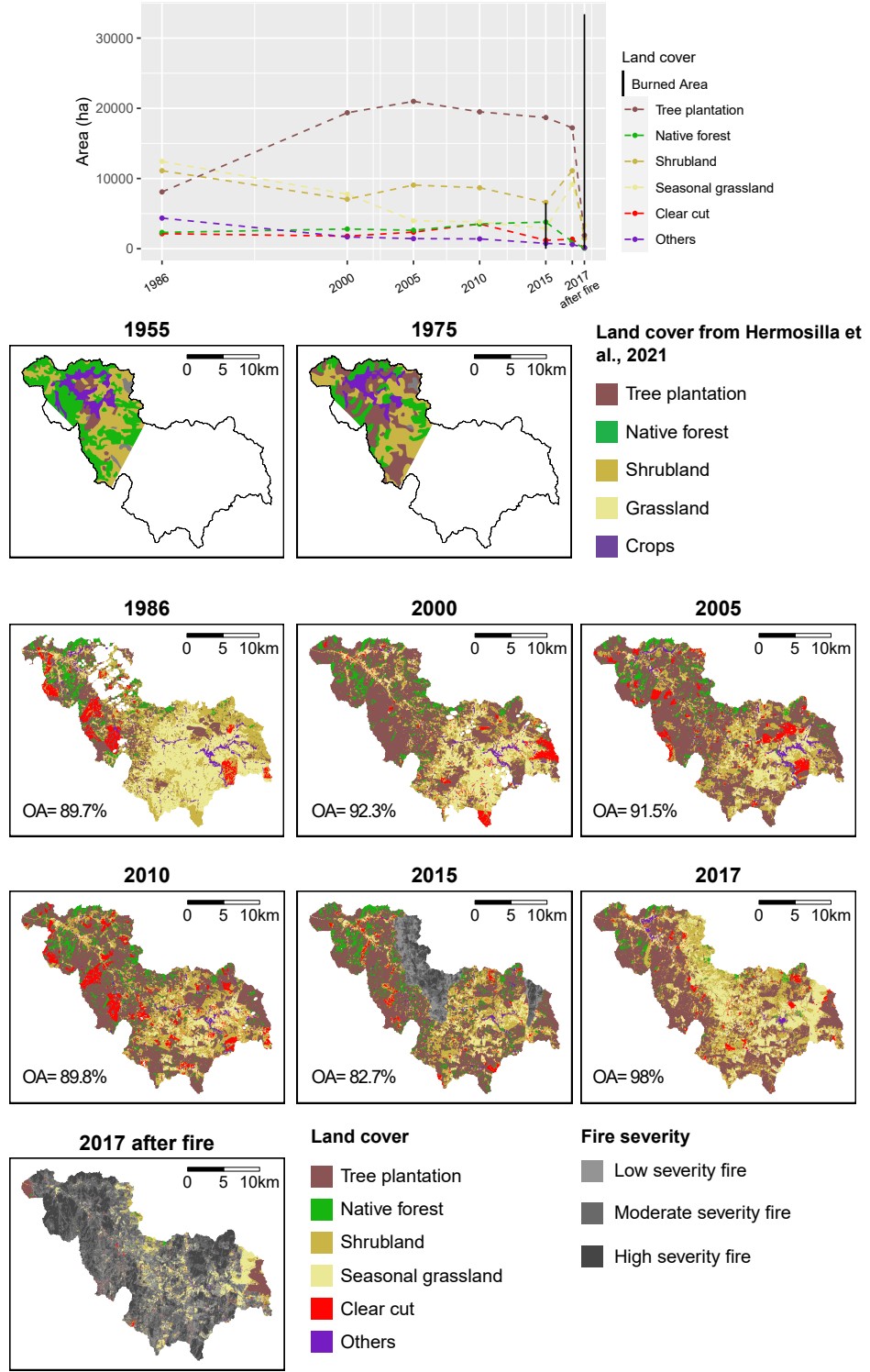

**Figure 7.** Land cover classification and transitions. Maps of 1955 and 1975 from Hermosilla-Palma et al. (2021), 1986-2015 from this work, and 2017 from Tolorza et al. (2022).

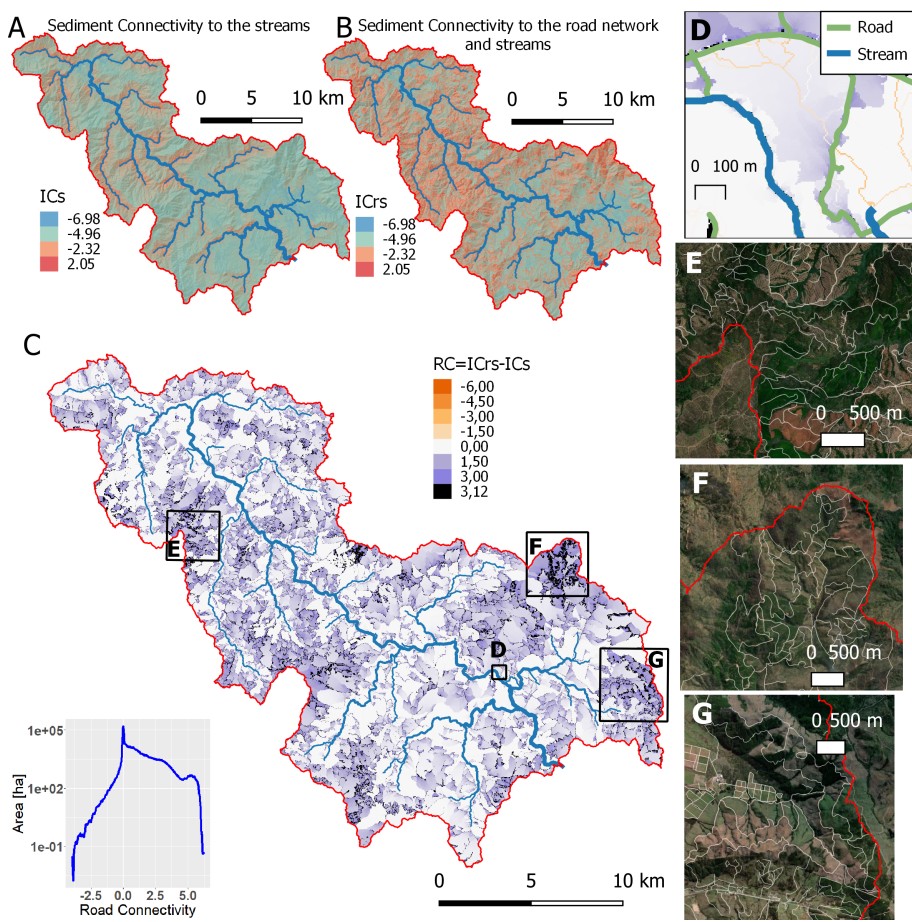

**Figure 8.** Sediment connectivity index (Cavalli et al., 2013) calculated using (A) the streams and (B) the streams and forest roads as targets. (C) Road Connectivity (RC) is the difference between both models. Histogram of RC is shown at the bottom left. Insets show location of maps D-G (D) Detail of RC as example of RC values using the same color table than C in relation to forestry roads and streams (E-G) Details of hilltops with highest values of $RC$ (©Google Maps 2022).

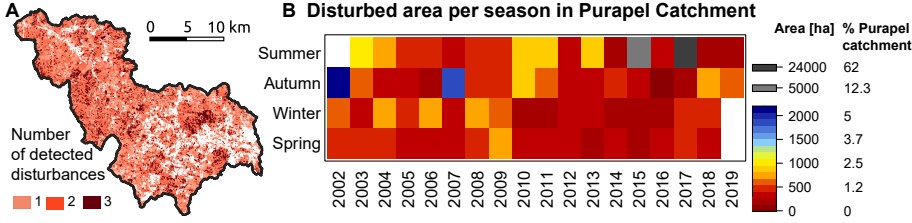

**Figure 9.** Detected disturbances from BFAST (A) map of the number of disturbances in vegetation detected for the period 2002-2019 (B) Seasonality of disturbance area detected within the Purapel catchment.