# Peer review of "Exotic tree plantations in the Chilean Coastal Range: Balancing effects of discrete disturbances, connectivity and a persistent drought on catchment erosion"

_EGUsphere, 2023_

## Author Response (AR1)

**RE: egusphere-2023-1783 (author) - final response "Exotic tree plantations in the Chilean Coastal Range: Balancing effects of discrete disturbances, connectivity and a persistent drought on catchment erosion"**

Dear Editorial Team of Earth Surface Dynamics,

Please find below our replies to the referee comments as requested. We thank you for considering our manuscript for publication in Earth Surface Dynamics. We particularly thank the Associate Editor Veerle Vanacker for handling our manuscript. We also express our thanks to Amanda Schmidt and Thomas Hoffmann for their time spent in providing constructive reviews and assessments. We highly appreciate all these efforts. In the following, we reply to the comments.

**Response to Reviewers**

**Referee Comment 1**

**Amanda Schmidt**

This paper looks at the effects of tree plantations on erosion in the coastal range in Chile by comparing suspended sediment concentration to in situ 10Be-derived erosion rates in the context of hillslope connectivity and detailed land use land cover change mapping. Although the authors do not have particularly conclusive results and the methods are not that new, I really like this paper. The data are well explained and the results are interesting. It's a bit puzzling to see so little difference between the two different metrics of erosion and the authors do a good job thinking about why that might be.

I do have a few very minor points that I would like to see the authors clarify.

We thank for the positive feedback. In the revised version of our manuscript we hopefully addressed all your concerns and suggestions where we found them applicable.

1) end of page 14 (around line 307), I got to wondering if storage in the system, like in floodplains or alluvial fans, could be part of the reason for the depressed sediment concentration. That is brought up later, but forecasting it earlier on would make things clearer.

We incorporate storage in this introductory paragraph of discussion. That paragraph now states:
```
[...]However, we regard the suspended-sediment decennial erosion rate as a very conservative
estimate for recent erosion.  First, we argue that this estimate does not account for possible
effects of sediment storage and other transient processes that affect soils and streams.  Second,
previous studies on rivers in the western Andes indicate that catchment-scale erosion from
gauging data may be underestimated due to under-sampling of extreme events (Vanacker et al.,
2020; Carretier et al., 2018).  Given the numerous data gaps of both streamflow and suspended
sediments, we cannot rule this out.  Also, we don't have sub-daily or depth-integrated measurements
of sediment concentrations.  Third, suspended sediments do not record the effects of chemical
weathering on denudation rates.[...]
```

2) The last word on line 307 ("This") is a pronoun that is unclear. I am not sure what "this" is.

We rephrase the related text (see second point above).

3) Is it possible that with the high rates of chemical weathering and likely deep regolith, the 10Be is an underestimate of long-term denudation rates? (see: Campbell, M. K., P. R. Bierman, A. H. Schmidt, R. S. Hernandez, A. Garcia-Moya, L. B. Corbett, A. J. Hidy, H. C. Aguila, A. G. Arruebarrena, G. Balco, D. Dethier, M. Caffee (2022). Cosmogenic nuclide and solute flux data from central Cuba emphasize the importance of both physical and chemical denudation in highly weathered landscapes. Geochronology. )

  4) Along the same lines of questioning the 10Be data, is it possible that you have stripped so much soil that these are artificially elevated and don't actually reflect the long-term pre-people rates? (like Hewawasam et al found in Sri Lanka [Hewawasam, Tilak, et al. "Increase of human over natural erosion rates in tropical highlands constrained by cosmogenic nuclides." Geology 31.7 (2003): 597-600.] or Schmidt et al (ESurf) found in China [Schmidt, A. H., Neilson, T. B., Bierman, P. R., Rood, D. H., Ouimet, W. B., and Sosa Gonzalez, V., 2016, Influence of topography and human activity on erosion in Yunnan: Earth Surface Dynamics , v. 4, n. 4, p. 819-830. http://www.earth-surf-dynam.net/4/819/2016/])

I agree that 10Be results may be affected both by intense weathering and the rapid soil erosion. Thanks very much for the literature and the ideas to improve our discussion on the long-term results:

    [...] This long-term rate, however, may even be overestimated: after 200 years of intense soil erosion, deep saprolite layers with low concentrations of $^{10}$Be are widely exposed at Earth Surface. Those hillslopes might be depleted in $^{10}$Be, which may lead to overestimate the total denudation (Schmidt et al., 2016)[...]

    [...] Indeed, deep chemical weathering may also lead to underestimating long-term denudation rates with $^{10}$Be. Such underestimation may occur if mass loss is high under the depth at which most cosmogenic nuclides are produced, as has been seen in tropical landscapes dominated by dissolution (Campbell et al., 2022)[...]

5) Is it possible that the hillslopes have been so disturbed that the sediment is totally stripped from them, leaving the hillslopes in a detachment limited system even while the valleys have stored sediment that is transport limited? Given the magnitude of erosion you are talking about, it seems like this could be possible. I could be entirely out to lunch though, not knowing the area, and I do think that intermediate hillslope storage is a reasonable explanation.

This landscape is probably transiting to a detachment limited system. Trees dig into soils where the inorganic component is each time coarser. We added some pictures of topsoil in Fig. 1 and the section 4.4 in the discussion to incorporate this idea.

6) The first two sentences of the conclusions are printed twice.

Corrected

This is a really neat study and I look forward to seeing the final version published.

We appreciate your review and evaluation. Your suggestions are quite pertinent and we hope you find better this version of the manuscript.

**Referee Comment 2**
**Thomas Hoffmann**

The authors present an interesting study on the development of stream flow and suspended sediment transport in Chilean headwater systems that are conditioned by changes of multiple drivers (drought, wild wire, tree plantations). This study is relevant and valuable to be published in the journal ESurf. The aim of the study, to extract the effect of tree plantations and wild fires on sediment transport is very challenging. In the end, the discussion of the results is very general and I have the feeling that the authors missed the chance to analyse the data in more detail (see general and detailed comments) and learn more about the specific controls. I made some suggestions to more specific approaches, which might shed more light to the discussion. Overall, I suggest major revisions following my general and detailed comments before publication.
Kind regards, Thomas

Thanks very much for your detailed review. We improve the discussion and respond here to your specific suggestions on data analysis.

General comments:
Unravelling causes/drivers of changing suspended sediment transport is a challenging task give the multiple interdepended drivers of suspended sediment loads. The authors are correct stating that an unambiguous attribution of cause and effect is difficult to assess. In this study the authors rely their statements mainly on the trend analysis. However, more statistical approaches are available to learn more about the driving factors.
The major issue is that SSD (load) is directly related to Q, since it is part of the calculation of the load. SSC (suspended sediment concentration) is strongly conditions by Q, but is not directly related to its estimation. Changes in sediment supply should therefore be discussed by changes in SSC. To see if changes in SSC are related to changes in Q or by changing sediment supply from hillslope sediment rating analysis could be performed, e.g. $SSC = a(Q/Qm)^b$) à changes in coefficient "a", which is the suspended sediment supply at Q=Qm should be related to changing supply conditions that are not related to changes in Q (see for instance Warrick 2015, WRR or Hoffmann et al. 2023, ESurf). I suggest to add a trend analysis of the rating parameters, to learn more about changing supplies.

We previously addressed the rating parameters of those data, although without the normalization by the geometric mean (Sauzal station in Figure 3, Tolorza et al., 2019). Then we knew about the large dispersion of SSC vs Q data in log-log space and we would anticipate a lack of trend for the intercept. Nonetheless, given your suggestion, we calculated the trend of the $\hat{a}$ intercept on $SSC = \hat{a}(Q/Qm)^b$, where Qm is the geometric mean. In the following plot we discard (1) hydrologic years with less than 185 daily data and (2)

years where the linear model of log transformed data resulted on pvalue > 0.05. The intercepts plotted in red are calculated for all the records and the intercepts plotted in blue are restricted to Q>Qm:

[Figure]

They do not present trend, according to the Mann-Kendall results. For that reason, we do not include those results in the MS.

Additionally, the observed patterns is superimposed by a declining trend. To extract the effect of single events (wild fire, Earthquake etc.) the authors should focus on the residuals of single years (with and without events) with repsect to these long-term trends.

We kindly thank you for raising this important issue. In fact, we discussed this previously and before submitting our initial manuscript. There are several points we want to emphasize here.

First, the fundamental question is how to estimate the baseline, i.e. the undisturbed state of the hydrological system to compare the measured data against. Here, a data-driven study, as the one we are performing, is limited, because we have to either include the data under 'disturbed' state or need to interpolate for the period of time when the disturbance may have effects. Both options are not adequate here, as both may introduce a bias. For example a regression plausibly (and most likely, too) experiences leverage effects if data from an disturbed state are included. The other option, i.e. interpolating for the questionable period of time, is neither feasible as we cannot exclude any side effects during the questionable time period, such as variable rainfall patterns driving the hydrological response. Accounting for all these limitations, we would need to perform a physics-based modeling exercise to quantify baselines, e.g. following Mohr et al. (2021). While we agree that a baseline level is important and the missing thereof definitely a weakness, we reject the suggestion here. Our study is data-driven and adding additional physics-based modeling exercises is out of the scope of our manuscript.

Detailed comments:
    Line 31: rephrase Mediterranean section (35-37.5o S) of the Chilean Coastal Range (CCR)"

Rephrased

Line 33: "slow denudation rates" à typically Mediterranean regions are characterized by high rates of soil erosion and gullying.

I agree that gullying and soil erosion under Mediterranean climate may be very high in terms of mass load, even more under anthropic intervention. However, in this paragraph we refer to the long-term and large-scale process of sculpting the gentle and largely convex hillslopes of (this specific portion of) the CCR. Here the slopes are much more gentle than the same Coastal Range in the Mediterranean latitudes where both the Coastal and the Andean range present their own peaks in denudation, (Carretier et al., 2018; Schaller et al., 2018)). It is expected that soil-mantled landscapes of gentle slopes denudate much more slowly than very steep landscapes dominated by landslides (e.g. Roering et al). This is in agreement with the observed detrital 10Be denudation rates on catchments draining the Andes Cordillera (landscapes dominated by landslides), which are orders of magnitude higher than denudation rates on the CCR (Carretier et al., 2018; Mohr et al., 2022).

Line 34: "secondary native forest" à should be explained

Done:
```
[...]Currently, the remnants of secondary native forest (i.e.  successional forests growing
in areas where the forest cover has at some time in the past been removed) stand on soils as
thick as 2 m[...]
```

Line 52: "the CCR ranks amongst the highest..." rephrase to "CCR ranks amongst regions with highest forest loss and gains worldwide".

rephrased

Line 55ff: are the statements limited to the "storm and yearly scale"? Furthermore, no other scales are mentioned later on à I suggest to remove the reference to the time scale.

Ok, we remove this reference to the time scale

Line 87: "transport-limited conditions" à this strongly depends on the dominant grain size. Only streams in the humid south of the CCR show transport limited conditions à see and refer to Terweh et al (2019, Geomorphology)

Thanks very much for pointing this study. We now referenced Terweh et al. (2021) in the discussion section. The related text was rephrased in the introduction.

Line 90: The two time-scales fall out off the box here: Why $10^4$ years? à You should motivate the use of the two time-scale. Since $10^4$ Yr is related to 10Be, we should explain here why you use 10Be!

Thanks very much for this suggestion. We move this explanation from the method section to the introduction.

[...]In most fluvial catchments, the long-term ($10^3 - 10^4$ years) denudation rates exceed short-term rates (Covault et al., 2013). This picture, however, may flip vice versa if soil erosion is high (Hewawasam et al., 2003; Vanacker et al., 2007). To evaluate this conundrum, we explore the catchment scale erosion and denudation of the Purapel river. To this end, we establish a long-term benchmark based on detrital $^{10}$Be denudation rate to compare recent sediment yields against. We include also discrete disturbance events[...]

Line 135: is there any chance to identify gaps due to ceased stream flow and gaps due to measurement errors/mistakes, defect measurement devices?

We couldn't unambiguously discriminate between ceased and unmeasured streamflow.

Line 149: the use of 10Be in the context of LULCC impacts should be motivated in the introduction

Done. See the text above in the response for line 90.

Line 213: The calculation of RC needs more explanation. How did you calculate IC_rs and IC_s? What was the reference point of both calculations? Does $IC_s$ is the connectivity of hillslopes with streams and $IC_{rs}$ the connectivity to either streams or roads, meaning that you treat roads as streams? What is the effect of subtracting both and what are the major assumptions in this approach? Please give more details on your approach. If I understood it correctly, calculating $IC_{rs}$ assumes that the connectivity of roads is 100%, meaning that very sediment that enters a road will immediately enter the stream. Am I correct and is this assumption valid? If yes, what about roads that are not connected to streams or that first go downhill and uphill afterwards? Could you show these effects of using $IC_s$ and $IC_{rs}$ based on a simplified graphic?

We incorporate more details in the description of methods and assumptions. In this case we consider the target as a temporary sink. Then, we are not assuming that sediments leaves immediately the catchment by entering the streams. Nevertheless, the routing of sediments is enhanced, since they leave more quickly their source area where the new target is closer downstream.

[...]The routing of sediments on hillslopes is likely to change where a new temporary sink is closer downslope. Both the streams and the forestry roads may behave as temporary sinks of sediments after their detachment on hillslopes (Schuller et al., 2013, 2021), and this behavior can be addressed using those elements (the streams and the forestry roads) as targets for $IC$. We propose here an approach to estimate the related changes in sediment connectivity by means of the difference between two different targets:

$$RC = IC_{rs} - IC_s \tag{1}$$

, where the subscripts of $IC$ refer to the target for its computation: $rs$ for the network formed by streams and roads and $s$ for the network formed only by the streams. We fed the model with a mapped forestry road network obtained from images available in the OpenLayers plugin

```
of QGIS and post-2017-fire Sentinel compositions.
    This approach accounts only for changes in sediment connectivity when forestry roads behave
as sinks of sediments, lacking their effect as sources.  [...]
```

Line 240ff: Not clear what exact criteria you used to define alpha=0.7. I would be helpful to use ranges of alpha to show confidence intervals of the analysis. The presented results of the main topic (changing suspended sediment loads) is very limited. More detailed description of the results should be given.

Changes in alpha do not produce a relevant change in Mann Kendall results for baseflow.

Line 255ff: This is in line with the comment in Fig. 5: It seems that you calculate the percentiles of annual estimates (as suggested by the units per year). Percentiles are derived from probability distributions, but as far as I understand you only have a single estimate of the annual load of each year. If you use the daily measurements as annual distributions you should highlight this using the correct units (e.g. t/day). Please clarify.

We include those units now in the plot.

Line 257: "Only the lower percentiles of SSD revert the decreasing trend at the end of the time series"?????? I don't see that!!!!!

p 0.25 (and maybe p50) after 2017 is higher than the period before the wildfires. This may explain the high p-value of this specific trend. I recognize this observation may be not clear, then I remove it from the text.

Line 260: How do you see if data are homoscedastic? Homoscedastic data show a normal distribution of residuals with respect to predicted values (e.g. using a linear regression). Why is homoscedasticity important in this context?

We applied the Fligner-Killeen test to evaluate if the variance of pre- and post-fire hydrometric data is or not constant. This is relevant to interpret the result of Wilcoxon-Mann-Whitney test, which is also explained in the method section.

Line 263: post-wild fire SSC/SSD is very low (due to low discharges) and changes of SSC are mainly related to changing discharges. Again, a rating analysis of pre and post-fire SSC Q relationships would be very helpful.

Post-fire $\hat{a}$ is within pre-fire variability, as can be observed in the plot above.

Line 276-277: you should also indicate whether there are places that show reduced connectivity, as the legend in Fig.8c indicates that there are areas with negative RC.

Now it is described the occurrence of negative RC both in the text on results section and with an histogram

in Fig 8.

Line 280: The threshold is very arbitrary and likely depends on the input data and their resolution. Therefore, it is questionable if the threshold from another study can be used here. If the definition of a "threshold" is necessary, why don't you use the distribution of IC to identify breakpoints.

This threshold is not arbitrary. It was found on a specific landscape by looking for quantitative estimations of connected and disconnected hillslopes. Reporting values over and below this threshold allows future comparisons between those different landscapes, while we also reported the area of higher RC value (surface where RC is over its p95) and the position within the landscape of that surface. We did that by reporting the contributing upstream area of the surfaces of high RC.

Line 328: In Figure 6 no mean stream flow is visible. The question is whether means stream flow is a good proxy of the impact of wildfire. Streamflow is strongly related to rainfall. Thus, stream flow should be related to rainfall to see the effects of the wild fire.

The Purapel catchment has a pluvial hydrological regimen, which is characterized by a quick and strong relationship between rainfall and streamflows. In Figure 6 comparisons of medians or means for different hydrologic years depend on the test used (Wilcoxon-Mann-Whitney or Welch t-test, respectively)

Line 335: Whether sediment transport is transport-limited or not depends on the grain size. This should be discussed.

We incorporate more discussion related to topsoil grain size and limitations in transport through time.

Line 338f: "a lack of minimum rainfall intensity required to trigger runoff and soil erosion on hillslopes (Mohr et al., 2013) and/or an increase in the residence time of sediments stored within the valleys is plausible." Not sure what you want to say?

We rephrased this text

Line 341-354: The discussion is very general and only weakly related to the results of the study. You should discuss the result from the IC here. The IC does not explain any changes of the transport capacity, but only regarding the connectivity between hillslopes and channels! How are the increased connectivity values related to the decreasing SSC?

We incorporate more discussion in the MS

Line 355ff: Again, limited links are drawn to the results of the study.

Discrete disturbances was widely described in the MS. Also, in the introduction the link with this discussion is explicit.

Figure 4: Stream flow and base flow show very similar pattern and similar tau values. Given the difficulties of defining the correct alpha value for stream flow separation and the similarity of the trends, I suggest to use total discharge as a more robust estimate of the discharge.

The decline in water storage is relevant for the discussion. We preserved this result in the MS

Figure 5: Please give more details on the percentiles. Percentiles of what? It seems that you calculate the percentiles of annual estimates. Percentiles are derived from probability distributions, but as far as I understand you only have a single estimate of the annual load of each year. I assume that you used the daily data here. This should somehow represent in the use of the unit. Please clarify.

We added your required units

Figure 6: Unfortunately there are major data gaps in SSC/SSD after the fires.

The fires are in the dry season. Those gaps probably are periods of zero streamflow.

Figure 7: I suggest to represent burned areas as bar and not as "*". The "*" are difficult to see.

Done

**References**

M. K. Campbell, P. R. Bierman, A. H. Schmidt, R. Sibello Hernández, A. García-Moya, L. B. Corbett, A. J. Hidy, H. Cartas Águila, A. Guillén Arruebarrena, G. Balco, D. Dethier, and M. Caffee. Cosmogenic nuclide and solute flux data from central Cuban rivers emphasize the importance of both physical and chemical mass loss from tropical landscapes. *Geochronology*, 4(2):435–453, 7 2022. ISSN 2628-3719. doi: 10.5194/gchron-4-435-2022. URL https://gchron.copernicus.org/articles/4/435/2022/.

S. Carretier, V. Tolorza, V. Regard, G. Aguilar, M. Bermúdez, J. Martinod, J.-L. Guyot, G. Hérail, and R. Riquelme. Review of erosion dynamics along the major N-S climatic gradient in Chile and perspectives. *Geomorphology*, 300:45–68, 1 2018. ISSN 0169555X. doi: 10.1016/j.geomorph.2017. 10.016. URL https://www.sciencedirect.com/science/article/pii/S0169555X17304506https://linkinghub.elsevier.com/retrieve/pii/S0169555X17304506.

J. A. Covault, W. H. Craddock, B. W. Romans, A. Fildani, and M. Gosai. Spatial and Temporal Variations in Landscape Evolution: Historic and Longer-Term Sediment Flux through Global Catchments. *The Journal of Geology*, 121(1):35–56, 1 2013. ISSN 00221376. doi: 10.1086/668680. URL http://www.jstor.org/stable/info/10.1086/668680.

T. Hewawasam, F. von Blanckenburg, M. Schaller, and P. Kubik. Increase of human over natural erosion rates in tropical highlands constrained by cosmogenic nuclides. *Geology*, 31(7):597–600, 2003. ISSN

00917613. doi: 10.1130/0091-7613(2003)031⟨0597:IOHONE⟩2.0.CO;2. URL `http://geology.gsapubs.org/content/31/7/597.abstract`.

C. Mohr, V. Tolorza, V. Georgieva, H. Munack, K. Wilcken, R. Fülöp, A. Codilean, E. Parra, and S. Carretier. Dense vegetation promotes denudation in Patagonian rainforests. *ESS OPEN ARCHIVE*, 2022. URL `https://essopenarchive.org/doi/full/10.1002/essoar.10511846.1`.

C. H. Mohr, M. Manga, G. Helle, I. Heinrich, L. Giese, and O. Korup. Trees Talk Tremor—Wood Anatomy and Content Reveal Contrasting Tree-Growth Responses to Earthquakes. *Journal of Geophysical Research: Biogeosciences*, 126(10), 10 2021. ISSN 2169-8953. doi: 10.1029/2021JG006385. URL `https://agupubs.onlinelibrary.wiley.com/doi/10.1029/2021JG006385`.

M. Schaller, T. A. Ehlers, K. A. Lang, M. Schmid, and J. P. Fuentes-Espoz. Addressing the contribution of climate and vegetation cover on hillslope denudation, Chilean Coastal Cordillera (26°–38°S). *Earth and Planetary Science Letters*, 489:111–122, 2018. ISSN 0012821X. doi: 10.1016/j.epsl.2018.02.026. URL `https://doi.org/10.1016/j.epsl.2018.02.026`.

A. H. Schmidt, T. B. Neilson, P. R. Bierman, D. H. Rood, W. B. Ouimet, and V. Sosa Gonzalez. Influence of topography and human activity on apparent in situ 10Be-derived erosion rates in Yunnan, SW China. *Earth Surface Dynamics*, 4(4):819–830, 11 2016. ISSN 2196-632X. doi: 10.5194/esurf-4-819-2016. URL `https://esurf.copernicus.org/articles/4/819/2016/`.

P. Schuller, D. E. Walling, A. Iroumé, C. Quilodrán, A. Castillo, and A. Navas. Using 137Cs and 210Pbex and other sediment source fingerprints to document suspended sediment sources in small forested catchments in south-central Chile. *Journal of Environmental Radioactivity*, 124:147–159, 2013. ISSN 0265931X. doi: 10.1016/j.jenvrad.2013.05.002. URL `http://dx.doi.org/10.1016/j.jenvrad.2013.05.002`.

P. Schuller, D. E. Walling, A. Iroumé, C. Quilodrán, and A. Castillo. Quantifying the temporal variation of the contribution of fine sediment sources to sediment yields from Chilean forested catchments during harvesting operations. *Bosque (Valdivia)*, 42(2):231–244, 2021. ISSN 0717-9200. doi: 10.4067/S0717-92002021000200231. URL `http://www.scielo.cl/scielo.php?script=sci_arttext&pid=S0717-92002021000200231&lng=en&nrm=iso&tlng=en`.

S. Terweh, M. A. Hassan, L. Mao, L. Schrott, and T. O. Hoffmann. Bio-climate affects hillslope and fluvial sediment grain size along the Chilean Coastal Cordillera. *Geomorphology*, 384:107700, 7 2021. ISSN 0169555X. doi: 10.1016/j.geomorph.2021.107700. URL `https://linkinghub.elsevier.com/retrieve/pii/S0169555X21001082`.

V. Tolorza, C. H. Mohr, S. Carretier, A. Serey, S. A. Sepúlveda, J. Tapia, and L. Pinto. Suspended Sediments in Chilean Rivers Reveal Low Postseismic Erosion After the Maule Earthquake (Mw 8.8) During a Severe Drought. *Journal of Geophysical Research: Earth Surface*, m:2018JF004766, 6 2019. ISSN 2169-9003. doi: 10.1029/2018JF004766. URL `https://onlinelibrary.wiley.com/doi/abs/10.1029/2018JF004766`.

V. Vanacker, F. von Blanckenburg, G. Govers, A. Molina, J. Poesen, J. Deckers, and P. Kubik. Restoring dense vegetation can slow mountain erosion to near natural benchmark levels. *Geology*, 35(4):303, 2007. ISSN 0091-7613. doi: 10.1130/G23109A.1. URL `http://geology.gsapubs.org/cgi/doi/10.1130/G23109A.1`.

V. Vanacker, M. Guns, F. Clapuyt, V. Balthazar, G. Tenorio, and A. Molina. Distribución espacio-temporal de los deslizamientos y erosión hídrica en una cuenca Andina tropical. *Pirineos*, 175:051, 9 2020. ISSN 1988-4281. doi: 10.3989/pirineos.2020.175001. URL `http://pirineos.revistas.csic.es/index.php/pirineos/article/view/310`.

---

## Author Response (AR2)

**RE: egusphere-2023-1783 (author) - final response "Exotic tree plantations in the Chilean Coastal Range: Balancing effects of discrete disturbances, connectivity and a persistent drought on catchment erosion"**

**Response to Reviewers**

**Report # 1**

**Thomas Hoffmann**

The quality of the manuscript greatly improved after the first revision and should be accepted after minor revisions. See comments below.

Kind regards Thomas Hoffmann

General comment:

The authors aim to identify the drivers of changings suspended sediment loads in the Purapel catchment, which has witnessed land use changes (tree plantations including strong clear-cuts), wild fires and superimposed climate changes (reduced rainfall which result in a reduction of discharge). The authors identify a declining trend of SSC esp during the autumn and winter months. Based on my first revision I suggested to add a sediment rating analysis to further analyze the reasons for the declining SSC. In their reply, the authors show that the rating coefficient a ($SSC = a(Q/Qgeom)^b$) shows not trend. This is e very interesting result which should be included in the manuscript because it highlights some potential explanations for changing SSC and SSL: SSC is changing but the rating coefficient "a" remains constant ("a" can be interpreted as SSC at Qgeom). This indicates that changes in SSC are not related to changes in sediment supply (which will affect "a"), but changes in discharge conditions (which affects average SSC, as SSC is related to Q). However, the authors argue not to include the rating analysis, because it does not show a significant trend. I suggest to include the analysis and to support the authors statement that SSL is reduced due to the sever drought conditions in the catchment.

Thanks for the positive feedback. We decided, however, to not include the SRC analysis. While our genuine work is out of this implicit context, we might open another completely new research thread. Results of that analysis my be interpreted in opposite behaviors, since the annual rating curves do indeed present some variability, where the exponent is slightly decreasing. Yet, the intercept does not show any trends:

[Figure]

In the previous plots, hydrologic years are represented only for years with >180 data points and with the pvalue of the lineal model of log transformed data <= 0.05. Interpreting these results in terms of sediment and water supply are challenging. A relatively constant $\hat{a}$ may result from a several scenarios, i.e. of changing SSD and Q alike (Fig 11 at Warrick, 2015). However, the rating curves are actually changing, and it's worth noting the trend in $b$. The observed decay of the exponent $b$ may indicate a lower dependency of SSC to Q. In turn, this may favor rather the hypothesis of supply limitation, just the contrary of your interpretation.

Detailed comments:
  Line 35+36: observing 2m thick soils under native forest, does not mean that this is the minimum soil thickness. It suggests that soils under natural vegetation cover might be 2m thick. However, soil thickness is strongly conditioned by topography (esp. curvature) as well. Please rephrase.

  Rephrased:
  ```
  Currently, the remnants of secondary native forests (i.e.  successional forests growing
in areas where forest cover was removed at some point in the past) stand on soils that are
up to 2 m (Soto et al., 2019), suggesting that soils under natural vegetation cover could reach
an even greater thickness.
  ```

Line 114: remove ")" after PET

Done

Line 254: "10Be denudation rater resulted in 0.024 ±0.004 mm/yr", rephrase

Done

Line 258: In addition to the limitations of the sampling for reliable load estimates, using SSD as a proxy of erosion rate is mainly limited by the sediment deliver ratio < 1.
We included that idea in the discussion section

Line 261: "Both rates do not statistically differ" −− > that is true, but does this assumption also hold if SDR of the catchments is <0.7 (as in many catchments of the world), which would increase the erosion rate?

  Now we described that drawback in the discussion section and estimated E=SSD/0.7

Line 293: "We achieved a confusion matrix with a balanced accuracy of 0.86 and a F1 score of 0.69" −− > this should be explained in the methods.

  Done. We add to the method section:
  ```
  The accuracy assessment was performed on 35 manually drawn polygons that were randomly distributed
across the catchment.  Since we were dealing with imbalanced data (looking for disturbances
  ```

which we assume as anomalies in the time series), we used balanced accuracy and F1 score as the metrics to evaluate our data (Brodersen et al., 2010). Balanced accuracy was calculated as the arithmetic mean of sensitivity and specificity while F1 score was calculated as

$$F1score = 2 * [(precision * recall)/(precision + recall)] \tag{1}$$

Line 317ff: you could clarify the transient storage effect using the SDR concept and state typical SDR values.

Done:

However, we regard the suspended-sediment decennial erosion rate as a very conservative estimate for recent catchment scale erosion for three reasons.

First, we argue that this estimate does not account for possible effects of sediment storage and other transient processes that affect soils and streams. For a certain time lapse, only a fraction of sediment detached within a catchment is transported to the outlet and, thus, captured in the sediment yield due to processes such as deposition and storage (Walling, 1983). The Sediment Delivery Ratio ($SDR$) describes that fraction, and use to be expressed as $SDR = SY/E$, where $SY$ is the average annual sediment yield per unit area and $E$ is the average annual erosion over that same area. Considering empirical data (Walling, 1983; Ferro and Minacapilli, 1995; Lu et al., 2006), a $SDR$ of 0.7 may be used as a conservative first approach for catchments larger than 100 km$^2$, although this value may be a couple of orders of magnitude lower. Using the mean specific $SSD$ as $SY$ in the previous expression, our estimation of decennial erosion at Purapel catchment increases to 67 $\pm$ 20 tkm$^{-2}$yr$^{-1}$ or 0.026 $\pm$ 0.007 mm/yr, which is even closer to the long-term denudation rate. The decennial catchment erosion may be much higher if the $SDR$ is orders of magnitude lower. More reliable estimates of $SDR$ can be done using spatially distributed models, such as the linear lumped series model of hillslope storage and channel storage of Lu et al. (2006). That model considers a theoretical relationship between sediment delivery and physical parameters such as particle size and rainfall duration.

Line 380: if you assume that tree plantations started 200 years ago, and that soils, which were 2m thick, are now removed, you would assume that average soil erosion during the last 200 years is in the order of 10mm/a, which is orders of magnitudes higher than rate presented in this study (mainly Chapter 3.2). If we assume that soil erosion adapted quickly after the introduction of tree plantations, then this erosion rate was likely much higher in the past and recent rates are now decreased due exhausted sediments (supply limitation). In this context, the similarity between long-term and recent rates can interpretd to be same erosion rate under supply limited conditions (but for different reasons) or be mere chance.

Thank you very much for this story. We added it to the discussions:

More than 200 years ago, the primary native forests probably grew on deep and carbon rich soils with surfaces probably much more enriched in fine sediments than the present topsoils. In contrast, the landscape-scale wildfire of 2017 revealed coarse saprolite widely exposed at the Earth surface (Fig. ??D). Roots of fallen trees digging up coarse sediments spread

out over most hillslopes. If we assume that landscape lowering has denuded 2 m of soil in
200 years, physical erosion should present one or more stages of much higher rates in the past
and after the beginning of intense deforestation. Rates of 10 mm/year are required to produce
such denudation in 200 years, which exceed both our short and long term estimates by 3 orders
of magnitude. While intense soil erosion occurs, the most likely evolution of grain size distribution
on topsoils is a quick depletion in fine sediments and a relative enrichment in coarse and
more weather resistant sediments. Together with deforestation and wildfires, this time-dependent
coarsening of the inorganic component of topsoils would be described as a reduction in the
supply of fine sediments from hillslopes to the streams, which may lead to a transition from
transport to supply limited catchment erosion. Because vegetation and soils have been systematically
disturbed, and given the high degree of coupling of those landscape elements and sediment mobilization
at streams, it is unlikely that the grain size distribution on rivers remain constant. Since
short-term and long-term erosion are comparable, we interpret that after a rapid response of
erosion to deforestation, denudation rates decreased due to exhausted sediments. Hence, it
is plausible that part of the observed decline in suspended sediment is related to changes
in the grain size distribution of sediments detached from soils. In this interpretation, both
the long- and the short- term erosion can be considered supply limited. The first due to the
scarcity of landslides and the second due to the low speed of soil (and fine sediment) production.

Line 386: strams −− > streams
Corrected

Line 388: than −− > that ?
Corrected

Line 390 +391: this is not supported by the results from the rating analysis. Changing SSC is mainly
related by decreased Q and not by decreased supply for a given discharge (i.e. Qgeom as expressed by the
a-coefficient).

We can interpret just the contrary from the negative trend of the exponent. But we are aware that several
different processes may produce similar results in the rating parameters. That's our main constrain to include
the rating analysis.

**Report # 2**
**Amanda Schmidt**

   I am satisfied that the authors adequately addressed the concerns that other reviewers and I made on
the first version of the manuscript.

We appreciate the contribution you made to our manuscript. Thanks very much for your positive feedback

**References**

K. H. Brodersen, C. S. Ong, K. E. Stephan, and J. M. Buhmann. The Balanced Accuracy and Its Posterior Distribution. In *2010 20th International Conference on Pattern Recognition*, pages 3121–3124. IEEE, 8 2010. ISBN 978-1-4244-7542-1. doi: 10.1109/ICPR.2010.764. URL http://ieeexplore.ieee.org/document/5597285/.

V. Ferro and M. Minacapilli. Sediment delivery processes at basin scale. *Hydrological Sciences Journal*, 40(6): 703–717, 12 1995. ISSN 0262-6667. doi: 10.1080/02626669509491460. URL http://www.tandfonline.com/doi/abs/10.1080/02626669509491460.

H. Lu, C. Moran, and I. P. Prosser. Modelling sediment delivery ratio over the Murray Darling Basin. *Environmental Modelling & Software*, 21(9):1297–1308, 9 2006. ISSN 13648152. doi: 10.1016/j.envsoft.2005.04.021. URL https://linkinghub.elsevier.com/retrieve/pii/S1364815205001386.

L. Soto, M. Galleguillos, O. Seguel, B. Sotomayor, and A. Lara. Assessment of soil physical properties' statuses under different land covers within a landscape dominated by exotic industrial tree plantations in south-central Chile. *Journal of Soil and Water Conservation*, 74(1):12–23, 12 2019. ISSN 0022-4561. doi: 10.2489/jswc.74.1.12. URL http://www.jswconline.org/lookup/doi/10.2489/jswc.74.1.12.

D. Walling. The sediment delivery problem. *Journal of Hydrology*, 65(1-3):209–237, 8 1983. ISSN 00221694. doi: 10.1016/0022-1694(83)90217-2. URL https://linkinghub.elsevier.com/retrieve/pii/0022169483902172.

J. A. Warrick. Trend analyses with river sediment rating curves. *Hydrological Processes*, 29(6):936–949, 3 2015. ISSN 0885-6087. doi: 10.1002/hyp.10198. URL https://onlinelibrary.wiley.com/doi/10.1002/hyp.10198.

---

## Author Response (AR3)

**RE: egusphere-2023-1783 (author) - manuscript accepted with corrections**

**Author's Response**

**Tom Coulthard**

**Comments to the author**

Dear Violeta and co-authors.

I am delighted to inform you that your paper has been accepted for final publication in ESurf - subject to some technical corrections. This is a minor editing stage before proof reading and with no need for any reviewing process - the AE has identified a few minor changes and it would be great if you could make these please. Its likely they would need to be done in the proof reading process itself, so this also saves time and effort there.

Finally, I would like to thank you for all your hard work with the paper and I look forward to seeing the final typset and formatted paper published!

All the best, Tom

Dear Editor,

We are very glad to have received acceptance of our manuscript. We have made the required modifications, which details are presented bellow.

**Associate editor decision: Publish subject to technical corrections**
**by Veerle Vanacker**

Public justification (visible to the public if the article is accepted and published):

This study from the Chilean Coastal Range nicely illustrates how anthropogenic disturbances (such as forest cover change or fires) alter catchment hydrology and sediment export. The authors used 10-Be derived denudation rates to benchmark modern erosion rates, and obtained modern erosion rates that are of the same order of magnitude as benchmark denudation rates. These results are unexpected and remarkable given the land use history of the region.

The revised manuscript nicely addressed the comments that were raised in the previous review rounds, and the revised discussion now provides more background on changes in the mechanisms of sediment production, transport, and export after forest plantations.

Veerle Vanacker

Thanks very much for all the improvements you made in our manuscript. We really appreciate your timely and dedicated review.

Additional private note (visible to authors and reviewers only):

A few minor suggestions

L17: hydroclimatic drivers

Corrected

Corrected

L97: as the timeseries is exceptionally long, you could state "over a 42-year period"

Modified

Figure 4 caption => C. Mean annual (?) suspended sediment concentration at....

Since the plot is showing the annual mean value of hydrometric data separated by seasons, we reworded the caption as follows:

```
    Mean seasonal streamflow, baseflow and suspended sediment concentrations at ''Purapel en
Sauzal'' station on an annual basis.  Main monotonic trends are tested with Mann-Kendall and
LOWESS smoothing for 1986-2018 (purple) and 2000-2018 (green).  Unfilled circles are discarded
data.  A. Mean seasonal streamflow at ''Purapel en Sauzal'' station.  B. Mean seasonal baseflow
at ''Purapel en Sauzal'' station.  C. Mean seasonal suspended sediment concentration at ''Purapel
en Sauzal'' station.
```

Corrected

L175: you chose '4 at/g/yr' for the SLHL production rate. Can you provide a reference here to justify the choice of 4 at/g/yr ?

Done:

```
    ...where P_{SLHL} = 4 at/g/yr is the sea-level-high-latitude total production rate of the
considered nuclide (Martin et al., 2017).  f_{sp}, f_{sm} and f_{fm} are the fractions of this production
rate due to spallation, slow muons capture and fast muons averaged over the catchment area,
respectively (Braucher et al., 2011).  S_{sp}, S_{sm}, S_{fm} are scaling factors depending on latitude
and elevation averaged over the catchment area (Stone, 2000), and ρ = 2.6 g/cm³.  No geometric
shielding correction for topography was applied (horizon < 20° in all directions).  The uncertainty
in the denudation rate is the propagation of the analytical uncertainty and an assumed uncertainty
of 15% in the production rate....
```

Table 1: Can you include the 10Be concentration that was measured in the blank in the caption of the table or in the text?

We incorporated this information in the supplementary table S1.

L241: Not entirely clear to me what you mean with "precision" and "recall" here. Can you add a sentence that briefly explains or describe how "precision" and "recall" are obtained, or what they represent?

We added the lines 239-240:

```
   Precision is the amount of correct positive predictions (true positives / (true positives
+ false positives)) and recall is how many positive predictions the model made over all positive
cases (true positives / (true positives + false negatives)).
```

L320 & 442: "at the earth surface" − > probably no need to capitalize the nouns here

modified

L383: "composed of" might be better here than "built by"

replaced

L413-415: this observation is very interesting, and similar to what was reported by Mediteranean basins in Spains where short-term and long-term erosion rates are also very similar (despite evidence of strong disturbances in the past such as e.g. during Roman times)

Thanks for this observation. It would be nice to have quantification of centennial-millenial erosion in those landscapes to better understand the evolution of degraded soils.

The CRN data are now summarized in Table 1. As they do not contain information on blanks, scaling and shielding factors, and production rates, they cannot easily be reused in further studies. Therefore, I suggest to add a table as supplement, or in Zenodo where all physical parameters are given that are needed for eventual reanalyses of the data. See e.g. Table 1 in Frankel et al. (2010) EOS 91(4).
   Reference. Frankel et al. (2010). Terrestrial Cosmogenic Nuclide Geochronology Data Reporting Standards Needed. Eos 91(4). https://agupubs.onlinelibrary.wiley.com/doi/pdf/10.1029/2010EO040003

Thanks very much for this observation. We included all the analytical information of our samples in the supplementary table 1.

**References**

R. Braucher, S. Merchel, J. Borgomano, and D. Bourlès. Production of cosmogenic radionuclides at great depth: A multi element approach. *Earth and Planetary Science Letters*, 309(1-2):1–9, 9 2011. ISSN 0012821X. doi: 10.1016/j.epsl.2011.06.036. URL https://linkinghub.elsevier.com/retrieve/pii/S0012821X11004079.

L. Martin, P.-H. Blard, G. Balco, J. Lavé, R. Delunel, N. Lifton, and V. Laurent. The CREp program and the ICE-D production rate calibration database: A fully parameterizable and updated online tool to compute

cosmic-ray exposure ages. *Quaternary Geochronology*, 38:25–49, 3 2017. ISSN 18711014. doi: 10.1016/j. quageo.2016.11.006. URL `https://linkinghub.elsevier.com/retrieve/pii/S1871101416300693`.

J. O. Stone. Air pressure and cosmogenic isotope production. *Journal of Geophysical Research*, 105 (B10):23753, 2000. ISSN 0148-0227. doi: 10.1029/2000JB900181. URL `http://dx.doi.org/10.1029/2000JB900181`.